# The Mediator complex regulates enhancer-promoter interactions

Shyam Ramasamy[1], Abrar Aljahani[1], Magdalena A. Karpinska [1],
T. B. Ngoc Cao [1], Taras Velychko [2], J. Neos Cruz[1], Michael Lidschreiber[2] &
A. Marieke Oudelaar [1]✉

Enhancer-mediated gene activation generally requires physical proximity between enhancers and their target gene promoters. However, the molecular mechanisms by which interactions between enhancers and promoters are formed are not well understood. Here, we investigate the function of the Mediator complex in the regulation of enhancer-promoter interactions, by combining rapid protein depletion and high-resolution MNase-based chromosome conformation capture approaches. We show that depletion of Mediator leads to reduced enhancer-promoter interaction frequencies, which are associated with a strong decrease in gene expression. In addition, we find increased interactions between CTCF-binding sites upon Mediator depletion. These changes in chromatin architecture are associated with a redistribution of the Cohesin complex on chromatin and a reduction in Cohesin occupancy at enhancers. Together, our results indicate that the Mediator and Cohesin complexes contribute to enhancer-promoter interactions and provide insights into the molecular mechanisms by which communication between enhancers and promoters is regulated.

Precise spatial and temporal patterns of gene expression in metazoans are regulated by enhancers, which are short non-coding DNA sequences that drive expression of their cognate gene promoters[1]. In mammals, enhancers can be located far upstream or downstream of the genes they control. To activate genes, enhancers interact with promoters in dynamic three-dimensional (3D) chromatin structures[2]. Enhancer-mediated gene activation is therefore closely related to the three-dimensional organization of the genome[3]. However, the molecular mechanisms by which enhancer-promoter interactions are formed and enhancers drive gene expression remain incompletely understood.

Mammalian genomes are organized into compartments and topologically associating domains (TADs). Compartments reflect separation of euchromatin and heterochromatin, whereas TADs represent relatively insulated regions of the genome, formed by loop extrusion[4]. In this process, ring-shaped Cohesin complexes translocate along chromatin and extrude progressively larger loops, until they are halted at CTCF-binding elements located at the boundaries of TADs[5]. Interacting enhancers and promoters are usually located in the same TAD[6]. Moreover, perturbations of TAD boundaries can cause ectopic enhancer-promoter interactions[7]. These observations suggest that loop extrusion could be involved in the regulation of enhancer-promoter communication and gene expression. Although it has been shown that depletion of components of the Cohesin complex does not lead to widespread mis-regulation of gene expression[8–10], Cohesin and its associated factors have been reported to be important for the regulation of cell-type-specific genes[11–13]. In addition, it has recently been shown that depletion of Cohesin can cause weakening of enhancer-promoter interactions[14,15]. These observations suggest that Cohesin-mediated loop extrusion contributes to the formation of enhancer-promoter interactions[16]. However, the molecular mechanism remains unclear. Furthermore, depletion of Cohesin causes a relatively subtle reduction in enhancer-promoter interaction strength[14]. This suggests that these interactions are not solely dependent on loop extrusion and that other mechanisms are involved in their formation.

Active enhancers and promoters are bound by transcription factors and coactivators, including the Mediator complex. Because the

[1]Genome Organization and Regulation, Max Planck Institute for Multidisciplinary Sciences, Göttingen, Germany. [2]Department of Molecular Biology, Max Planck Institute for Multidisciplinary Sciences, Göttingen, Germany. ✉e-mail: marieke.oudelaar@mpinat.mpg.de

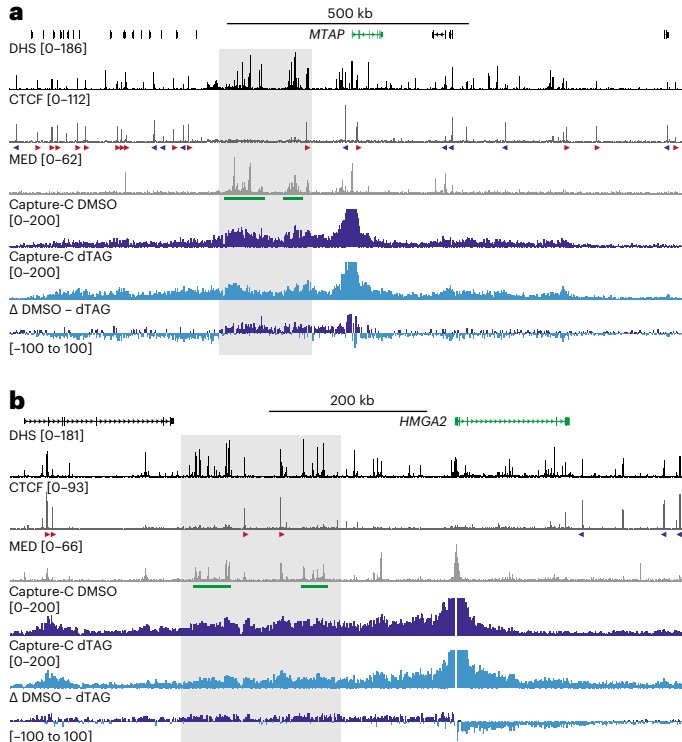

**Fig. 1 | Changes in chromatin interactions upon Mediator depletion.**
**a**, Capture-C interaction profiles from the viewpoint of the *MTAP* promoter in
HCT-116 MED14-dTAG cells treated with DMSO (dark blue; *n* = 3 biologically
independent samples) or dTAG ligand (light blue; *n* = 3 biologically independent
samples). Gene annotation, DNase-hypersensitive sites (DHS) and ChIP–seq
data for CTCF and MED26 are shown above and a differential profile (Δ DMSO
– dTAG) is shown below. Super-enhancers are highlighted in green below the
MED26 profiles and orientations of CTCF motifs are indicated with arrowheads
(forward orientation in red; reverse orientation in blue). The gray box highlights
a broad reduction in interactions in the region containing super-enhancers
in dTAG-treated cells. The axes of the DHS and ChIP–seq profiles are scaled to
signal; the axes of the Capture-C profiles are fixed (ranges indicated in brackets).
Coordinates (hg38): chr9:21,096,000–22,491,000. **b**, Data are as described in
**a**, but for the *HMGA2* locus. Coordinates (hg38): chr12:65,260,000–66,115,000.

tail module of the Mediator complex interacts with the activation
domains of transcription factors bound at enhancers and the head
and middle modules interact with the pre-initiation complex (PIC) at
gene promoters[17–19], it has been proposed that Mediator acts as a bridge
between enhancers and promoters (reviewed in refs. 20–23). Initial
studies based on knockdown of Mediator subunits over the course
of several days provided evidence for this hypothesis[24–26]. However,
since the Mediator complex has a central function in RNA polymerase
II (Pol II)-mediated transcription, its long-term perturbation causes
secondary, confounding effects, which complicate the interpretation
of these early studies.

To overcome these limitations, more recent studies have used
rapid protein-depletion strategies to investigate the function of the
Mediator complex in gene regulation and genome organization[27,28].
These studies did not detect changes in chromatin architecture and
enhancer-promoter interactions upon Mediator depletion, despite
strongly reduced expression levels of enhancer-dependent genes. On the
basis of these findings, it has been concluded that Mediator is dispensa-
ble for enhancer-promoter interactions and acts as a functional rather
than an architectural bridge between enhancers and promoters[27,28].

A caveat of current studies of the role of Mediator in genome archi-
tecture is that enhancer-promoter interactions have been assessed with
chromosome conformation capture (3C) methods at relatively low

resolution[24–29]. It is therefore possible that changes in fine-scale genome
architecture, including enhancer-promoter interactions, could not be
reliably identified. For a better understanding of the function of the
Mediator complex in genome regulation, it is important to examine
the impact of acute Mediator perturbations on chromatin architecture
with high resolution and sensitivity.

Here, we overcome limitations of current studies and investigate
the function of the Mediator complex by combining rapid protein
depletion and high-resolution analysis of genome architecture using
both conventional and MNase-based 3C approaches. We find that deple-
tion of Mediator leads to a significant reduction of enhancer-promoter
interactions. Interestingly, we also find that Mediator depletion causes
increased interactions between CTCF-binding elements. We show that
these changes in interaction patterns are associated with a redistribu-
tion of the Cohesin complex on chromatin and a loss of Cohesin occu-
pancy at enhancers upon Mediator depletion. These results suggest
that enhancer-promoter interactions are dependent on both Mediator
and Cohesin and provide support for a model in which the Cohesin
complex bridges and stabilizes interactions between enhancers and
promoters bound by Mediator.

## Results

### Mediator depletion causes changes in chromatin interactions
Because the MED14 subunit acts as a central backbone that connects
the Mediator head, middle and tail modules[19], its degradation disrupts
the integrity of the Mediator complex[27,28]. We have therefore used an
HCT-116 MED14-dTAG cell line[28] to study the function of the Mediator
complex in genome regulation. Using immunoblotting (Extended
Data Fig. 1a,b) and chromatin immunoprecipitation and sequencing
(ChIP–seq) (Extended Data Fig. 1c–i), we have confirmed efficient
MED14 depletion within 2 h of treatment with a dTAG ligand.

Previous work has shown that Mediator depletion leads to strong
downregulation of cell-type-specific genes that are associated with
super-enhancers[28] (Extended Data Fig. 1j). Super-enhancers are
stretches of clustered enhancers with high levels of Mediator that are
thought to have a central role in driving high expression levels of key cell
identity genes[30]. Previous studies could not detect changes in interac-
tions between promoters and (super-)enhancers upon Mediator deple-
tion[27–29]. However, these studies relied on genome-wide 3C approaches,
such as Hi-C and Hi-ChIP, with relatively low resolution (4–5 kb). It is
therefore possible that small-scale changes in enhancer-promoter
interactions could not be reliably detected.

To investigate changes in genome architecture upon Mediator
depletion in more detail, we used targeted 3C approaches, which are
not limited by sequencing depth and can detect changes in genome
structure at high resolution and with high sensitivity. We focused our
analyses on 20 genes (Extended Data Fig. 1j), which we selected on the
basis of the following criteria: (1) robust gene activity in HCT-116 cells;
(2) significant downregulation of gene expression upon Mediator
depletion; (3) high Mediator occupancy at the gene promoter; and (4)
association with a super-enhancer. We initially used Capture-C[31,32], a
targeted 3C method based on DpnII digestion, to evaluate changes in
chromatin interactions with the promoters of these genes. Capture-C
interaction profiles display interaction frequencies with selected view-
points per DpnII restriction fragment and therefore have an average
resolution of ~250 bp. By comparing Capture-C data generated in HCT-
116 MED14-dTAG cells treated with DMSO or dTAG ligand, we find that
Mediator depletion leads to subtle changes in the interaction patterns
of the selected gene promoters (Fig. 1 and Extended Data Fig. 2a–d).
Unexpectedly, we observe patterns of both decreased and increased
interactions.

For example, in the *MTAP* locus, the Capture-C data show reduced
interactions in the upstream region, in which two super-enhancers
are located, and a trend towards increased interactions in the regions
further upstream and downstream (Fig. 1a). In the region containing

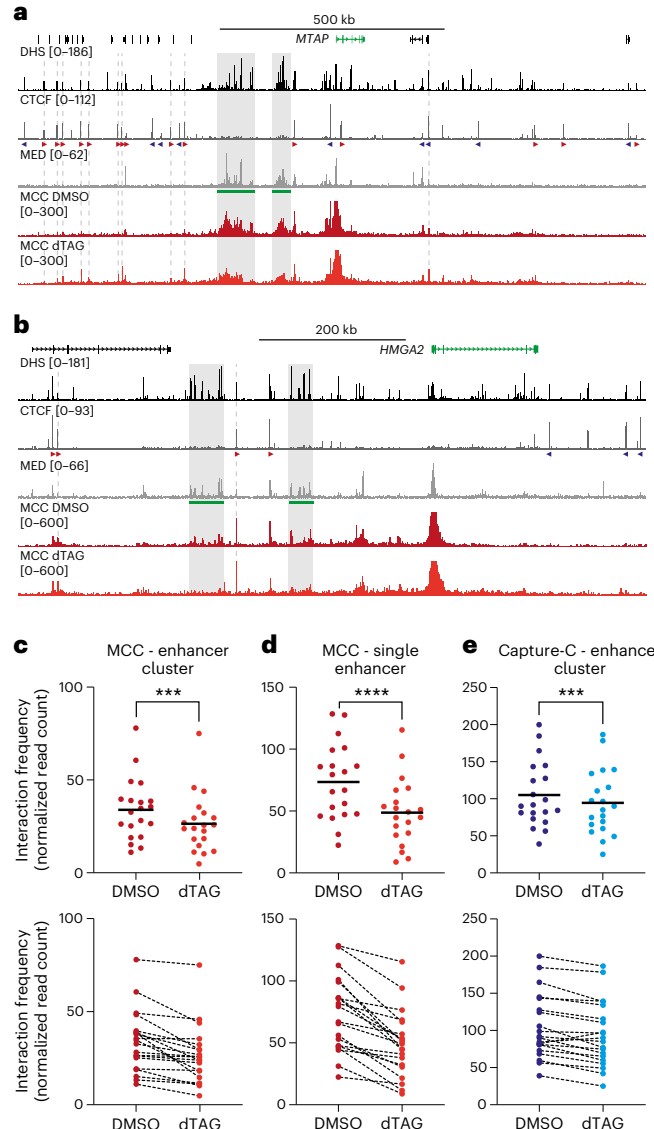

**Fig. 2 | Depletion of Mediator leads to decreased enhancer-promoter interactions and increased interactions with CTCF-binding sites.**
**a**, Micro-Capture-C (MCC) interaction profiles from the viewpoint of the *MTAP* promoter in HCT-116 MED14-dTAG cells treated with DMSO (dark red; n = 3 biologically independent samples) or dTAG ligand (light red; n = 3 biologically independent samples). Gene annotation, DHS and ChIP–seq data for CTCF and MED26 are shown above. Super-enhancers are highlighted in green below the MED26 profiles, and orientations of CTCF motifs are indicated with arrowheads (forward orientation in red; reverse orientation in blue). The gray boxes highlight reduced interactions between the promoter and super-enhancers; the gray dashed lines highlight increased interactions with CTCF-binding sites. The axes of the DHS and ChIP–seq profiles are scaled to signal; the axes of the MCC profiles are fixed (ranges indicated in brackets). Coordinates (hg38): chr9:21,096,000–22,491,000.
**b**, Data are as described in **a**, but for the *HMGA2* locus. Coordinates (hg38): chr12:65,260,000–66,115,000. **c**, Quantification of interaction frequencies between gene promoters and enhancer clusters (average size: 58 kb), extracted from MCC data in 20 loci. ***P = 0.000162 (two-sided ratio paired *t*-test). **d**, Quantification of interaction frequencies between gene promoters and individual enhancers (average size: 2.7 kb), extracted from MCC data in 20 loci. ****P = 0.000011 (two-sided ratio paired *t*-test). **e**, Quantification of interaction frequencies between gene promoters and enhancer clusters (average size: 58 kb) extracted from the Capture-C data presented in Fig. 1 in 20 loci. ***P = 0.000628 (two-sided ratio paired *t*-test).

the *HMGA2* oncogene, there are fewer interactions in the upstream region, which contains two super-enhancers, whereas interactions in the downstream region are increased (Fig. 1b). We observe similar

patterns in other loci we investigated (Extended Data Fig. 2a–d). However, in regions containing genes that are not highly expressed in HCT-116 cells and are not sensitive to Mediator depletion, we do not see clear changes in interaction patterns (Extended Data Fig. 2e,f).

## Depletion of Mediator reduces enhancer-promoter interactions

To examine the broad changes in the Capture-C interaction profiles in further detail, we performed Micro-Capture-C (MCC) experiments[33] in DMSO- and dTAG-treated HCT-116 MED14-dTAG cells, using viewpoints targeting the same set of gene promoters. Compared with Capture-C, MCC has an advantage in that it uses MNase instead of DpnII for chromatin digestion. The resolution of MCC is therefore not limited by the distribution of DpnII cut sites across the genome, enabling analysis at base-pair resolution[33]. The MCC data resolve the broad interaction patterns in the Capture-C data and clearly show that Mediator depletion leads to reduced interactions between gene promoters and Mediator-bound enhancer regions in the *MTAP* and *HMGA2* loci (Fig. 2a,b).

We find that depletion of Mediator leads to a decrease in the frequency of enhancer-promoter interactions in the 20 regions that we focused on (Extended Data Fig. 2g–j). Quantification of the MCC interactions between gene promoters and clusters of Mediator-bound enhancers indicates an average reduction of 22% across these regions (Fig. 2c). The reduction in interaction frequency between the promoters and a narrow region covering the largest Mediator peak within these broad clusters is, on average, 34% (Fig. 2d). These changes are associated with an average decrease in gene expression of 7.5-fold (Extended Data Fig. 1j). Of note, the Capture-C data also detect a reduction in enhancer-promoter interactions in most regions of interest, with an average decrease in interaction frequency of 9% (Fig. 2e). Although it is in accordance with the MCC data, this comparison highlights the need for analyses with sufficient resolution and sensitivity to robustly detect changes in enhancer-promoter interactions.

## CTCF-dependent interactions increase upon Mediator depletion

The MCC data do not only identify specific reductions in interactions with enhancers, but also uncover very precise increased interactions following depletion of Mediator. Strikingly, these increased interactions all overlap with CTCF-binding sites. For example, in the CTCF-dense *MTAP* locus, we see strong increases in interactions formed with CTCF-binding sites in the region upstream of the super-enhancers and downstream of the gene promoter (Fig. 2a). Notably, the interacting CTCF-binding sites upstream are all in a forward orientation, whereas the interacting CTCF-binding sites downstream are all in a reverse orientation.

We observe a similar pattern of increased interactions with convergently orientated CTCF-binding sites in the *MYC* locus after Mediator depletion (Extended Data Fig. 2g). In the *HMGA2*, *ITPRID2*, *ERRFI1* and *KRT19* loci, which contain fewer CTCF-binding sites, the patterns are a bit more subtle, but also clearly present (Fig. 2b and Extended Data Fig. 2h–j).

It has been suggested that MNase-based 3C data could be biased by varying chromatin accessibility and MNase digestion efficiency across regions or conditions. However, the fact that we detect a significant decrease in enhancer-promoter interactions in both the MCC and the Capture-C data, which are generated with restriction enzyme digestion, indicates that reduced enhancer-promoter interactions after depletion of Mediator are unlikely to reflect underlying changes in chromatin accessibility. In addition, the observation of both decreased and increased interactions following depletion of Mediator, with increased interactions specifically overlapping with CTCF-binding sites in a convergent orientation, indicates that it is improbable that the MCC data are skewed by nucleosome positioning. To further demonstrate that the

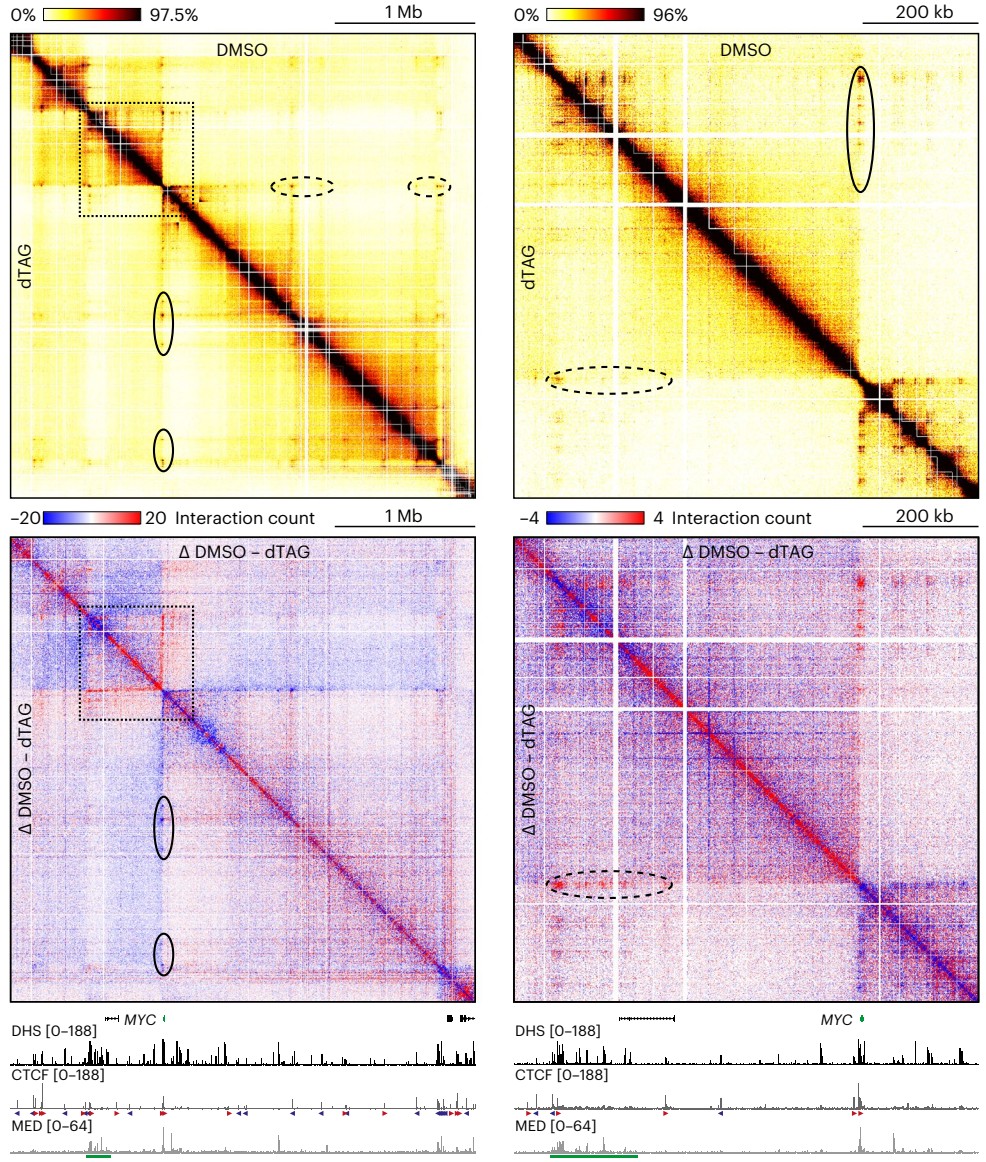

**Fig. 3 | Mediator depletion results in subtle changes in large-scale genome organization.** Tiled-MCC contact matrices of the *MYC* locus in HCT-116 MED14-dTAG cells treated with DMSO (top right; *n* = 3 biologically independent samples) or dTAG ligand (bottom left; *n* = 3 biologically independent samples). The matrices on the right show a zoomed view of the area enclosed by the dashed squares in the left matrices. Differential contact matrices, in which interactions enriched in DMSO-treated cells are shown in red and interactions enriched in dTAG-treated cells are shown in blue, are displayed below. Gene annotation,

DHS and ChIP–seq data for CTCF and MED26 are shown at the bottom. Super-enhancers are highlighted in green below the MED26 profiles, and orientations of CTCF motifs are indicated with arrowheads (forward orientation in red; reverse orientation in blue). The dashed black ovals in the dTAG and differential contact matrices highlight decreased enhancer-promoter interactions, whereas the solid ovals indicate increased CTCF interactions. Coordinates (hg38): chr8:126,650,000–129,950,000.

changes in chromatin interactions in Mediator-depleted cells are not biased by potential changes in accessibility affecting MNase digestion, we performed ATAC-seq experiments[34] in DMSO- and dTAG-treated HCT-116 MED14-dTAG cells (Extended Data Fig. 3). These experiments show that Mediator depletion does not lead to strong changes in chromatin accessibility in our regions of interest. Together, these observations indicate that the changes in enhancer-promoter interactions detected by MCC reflect bona fide changes in chromatin architecture.

**Mediator depletion causes changes in intra-TAD interactions**
The MCC data show clear and precise changes in chromatin interactions upon depletion of the Mediator complex. However, since the MCC viewpoints are very narrow and focused on gene promoters, it remains unclear how large-scale 3D genome architecture is changed, and how

interactions between other *cis*-regulatory elements are impacted by Mediator depletion. We therefore used the Tiled-MCC approach, in which MCC library preparation is combined with an enrichment strategy based on capture oligonucleotides tiled across large genomic regions of interest[14], to investigate changes in genome architecture in DMSO- and dTAG-treated HCT-116 MED14-dTAG cells in a broader context. We focused on the *MYC* (3.3 Mb; Fig. 3), *MTAP* (1.55 Mb; Extended Data Fig. 4), *HMGA2* (990 kb; Extended Data Fig. 5) and *ITPRID2* (900 kb; Extended Data Fig. 6) loci.

In line with previous studies that have used Hi-C or Hi-ChIP to examine changes in genome architecture[27–29], we do not detect drastic changes in large-scale genome organization after Mediator depletion. We find that TAD organization is preserved, without any shifts in the location of boundaries. However, we find subtle changes in interaction

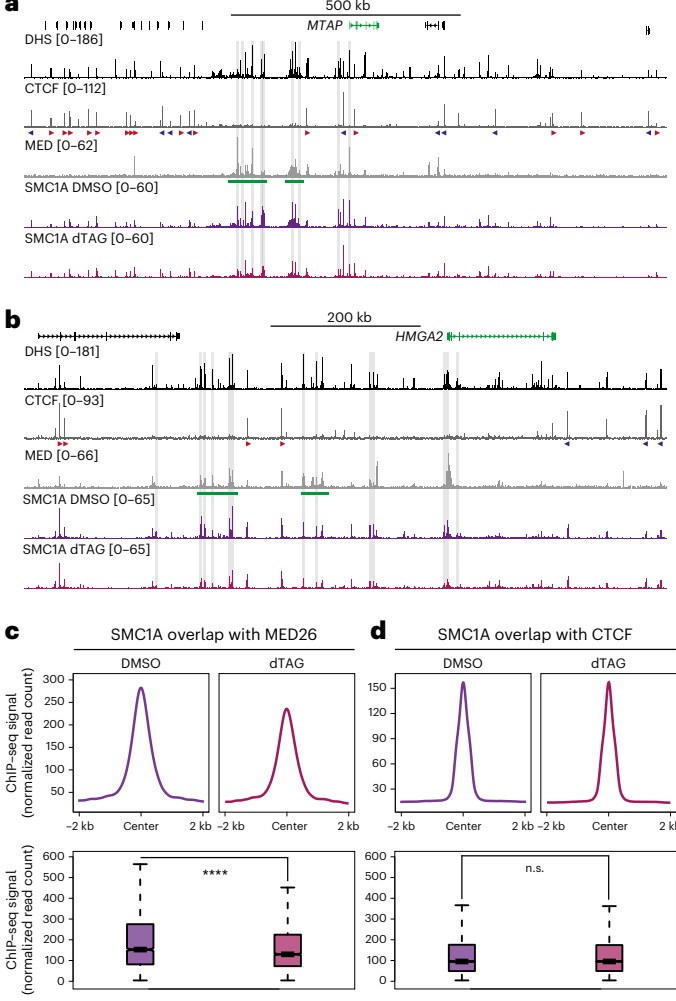

**Fig. 4 | Cohesin occupancy at enhancers is reduced after depletion of Mediator. a,** CUT&Tag data for the Cohesin subunit SMC1A in the *MTAP* locus in HCT-116 MED14-dTAG cells treated with DMSO (dark purple; *n* = 3 biologically independent samples) or dTAG ligand (light purple; *n* = 3 biologically independent samples). Gene annotation, DHS and ChIP–seq data for CTCF and MED26 are shown above. Super-enhancers are highlighted in green below the MED26 profiles, and orientations of CTCF motifs are indicated with arrowheads (forward orientation in red; reverse orientation in blue). The gray bars highlight SMC1A peaks that are significantly reduced after Mediator depletion (*P* < 0.05; Supplementary Table 1). The axes of the DHS and ChIP–seq profiles are scaled to signal; the axes of the CUT&Tag profiles are fixed (ranges indicated in brackets). Coordinates (hg38): chr9:21,096,000–22,491,000. **b,** Data are as described in **a,** but for the *HMGA2* locus. Coordinates (hg38): chr12:65,260,000–66,115,000. **c,** Meta-analysis of SMC1A peaks overlapping with MED26 peaks in HCT-116 MED14-dTAG cells treated with DMSO (dark purple) or dTAG ligand (light purple). The box plot shows the median and the interquartile range (IQR) of the data, and the whiskers indicate 1.5 × IQR values. ****$P$ = 3.578 × 10$^{-13}$ (two-sided Wilcoxon rank sum test). **d,** Data are as described in **c,** but for SMC1A peaks overlapping with CTCF. n.s., not significant (*P* = 0.8174; two-sided Wilcoxon rank sum test).

patterns within TADs. In line with the Capture-C and MCC data, we observe that enhancer-promoter interactions are reduced after Mediator depletion. In addition, we detect strengthening of interactions anchored at CTCF-binding sites. As a result, we see subtle increases in 'looping' between the CTCF-bound anchors of TADs and sub-TADs.

**Cohesin binding patterns are altered upon Mediator depletion**
It has been shown that CTCF and Cohesin co-localize and that interactions between CTCF-binding sites are formed via loop extrusion by the

Cohesin complex[35–38]. Notably, Cohesin also co-localizes with Mediator, and co-immunoprecipitation experiments have suggested that these complexes interact[24,25,39]. However, a functional link between Mediator and Cohesin has not been identified.

Because our data show that depletion of Mediator causes a decrease in enhancer-promoter interactions and an increase in CTCF-mediated interactions, we hypothesized that these altered interaction patterns could be explained by changes in the distribution of the Cohesin complex on chromatin. To test this, we mapped Cohesin occupancy using cleavage under targets and tagmentation (CUT&Tag[40]) in DMSO- and dTAG-treated HCT-116 MED14-dTAG cells (Fig. 4 and Extended Data Fig. 7). These data show clear changes in Cohesin occupancy upon Mediator depletion. For example, in the *MTAP* and *HMGA2* loci, we observe a significant reduction in Cohesin levels at the super-enhancers and other Mediator-bound elements (Fig. 4a,b). By contrast, Cohesin occupancy at CTCF-binding sites in these regions is not grossly affected by Mediator depletion. We find similar patterns in the *MYC*, *ITPRID2*, *ERRFI1* and *KRT19* loci (Extended Data Fig. 7). Genome-wide quantification of Cohesin occupancy at Mediator-bound enhancers and CTCF-binding sites shows a significant reduction in Cohesin levels at enhancers and stable occupancy at CTCF-binding sites after depletion of Mediator (Fig. 4c,d). These results show that the distribution of Cohesin is altered when Mediator is depleted and suggest that Mediator contributes to the stabilization of Cohesin at enhancer elements.

**Mediator depletion causes changes in nano-scale interactions**
To further analyze the impact of Mediator depletion on chromatin architecture, we leveraged the ability of Tiled-MCC to directly identify ligation junctions and resolve localized nano-scale interaction patterns[14]. We focused our analyses on ligation junctions in regions containing super-enhancers, genes and boundary elements in the *MYC*, *MTAP*, *HMGA2* and *ITPRID2* loci (Fig. 5 and Extended Data Fig. 8).

Within the *MYC* super-enhancer, we observe enriched interactions between the individual elements of the super-enhancer (Fig. 5, left matrix). After depletion of Mediator, the frequency of these interactions is decreased. We observe similar patterns in the *MTAP*, *HMGA2* and *ITPRID2* loci (Extended Data Fig. 8). In the *MTAP* locus, we could also resolve the interactions between the gene promoter and a nearby enhancer. After Mediator depletion, there are fewer of these interactions (Extended Data Fig. 8a). These results show that interactions between active enhancer and promoter elements across very small distances are dependent on Mediator.

It has previously been shown that regions containing CTCF-binding sites form characteristic architectural patterns, in which phased nucleosomes surrounding the CTCF motif form a grid-like structure, which is associated with strong insulation between the regions upstream and downstream of the CTCF-binding site[14,41]. We observe these patterns at the intergenic CTCF-binding sites in the loci we investigated and do not see any changes upon depletion of Mediator (Fig. 5, right matrix, and Extended Data Fig. 8).

At the level of individual genes, we observe domain-like structures extending across the gene body (Fig. 5, middle-left matrix, and Extended Data Fig. 8). Interestingly, we observe that depletion of Mediator results in the appearance of specific structures within the *MYC* gene, which are centered around hypersensitive and CTCF-bound elements (Fig. 5, middle-left matrix). Zooming in on this region at higher resolution (Fig. 5, middle-right matrix) resolves a structure that is reminiscent of intergenic CTCF-binding sites at the CTCF-bound region within the *MYC* gene body when Mediator is depleted. This suggests that high transcriptional activity in the presence of Mediator leads to a disruption of the specific nucleosome structures that are normally formed around CTCF-binding sites. We observe similar patterns at the CTCF-binding sites contained within the *MTAP* and *ITPRID2* gene bodies upon depletion of Mediator (Extended Data Fig. 8a,c).

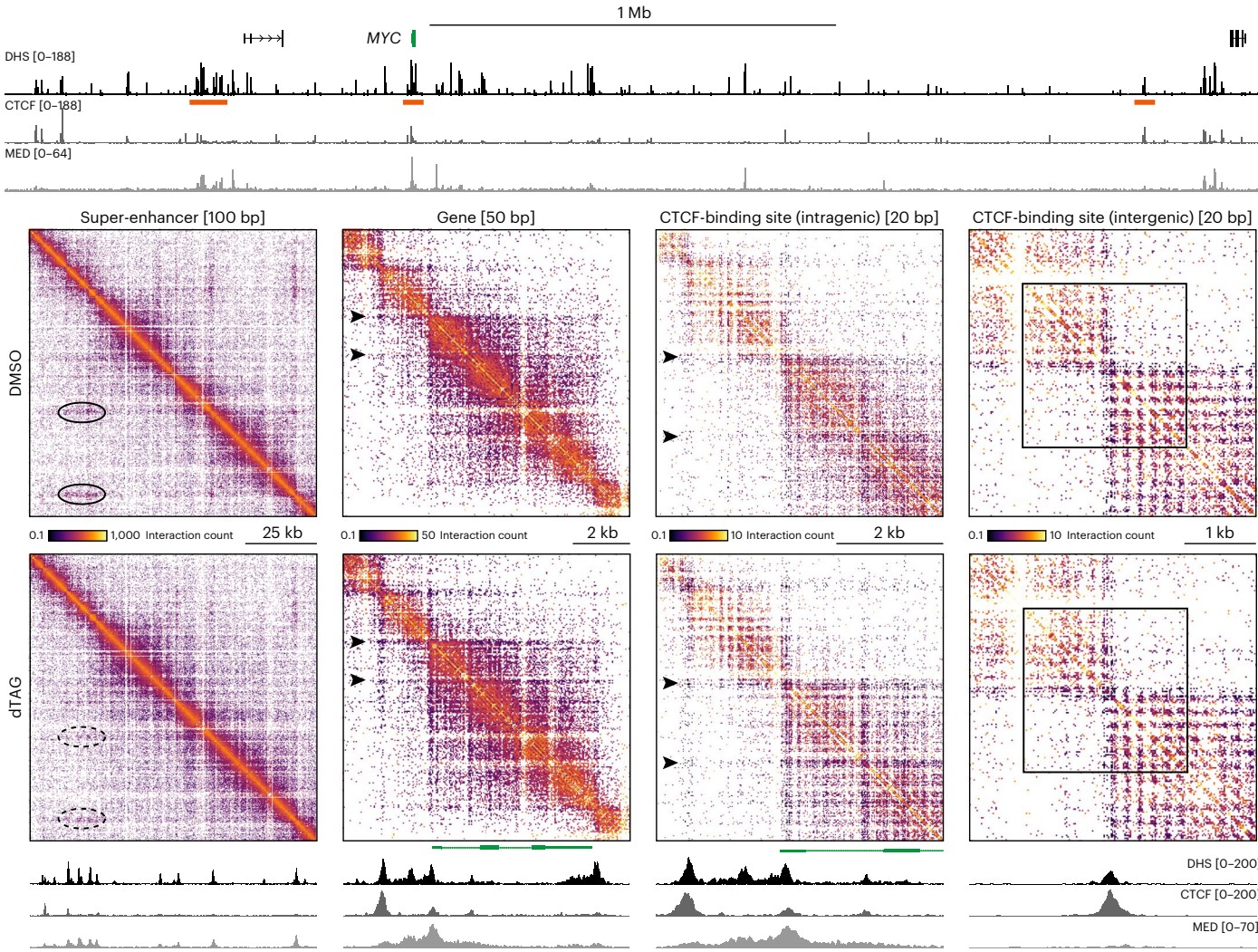

**Fig. 5 | Depletion of Mediator leads to changes in nano-scale genome organization.** Tiled-MCC ligation junctions in the *MYC* locus in HCT-116 MED14-dTAG cells treated with DMSO (top; *n* = 3 biologically independent samples) or dTAG ligand (bottom; *n* = 3 biologically independent samples), displayed in localized contact matrices at high resolution. Gene annotation, DHS and ChIP–seq data for CTCF and MED26 for the extended and localized *MYC* locus are shown above and below the matrices, respectively. The regions covered in the contact matrices are highlighted with orange bars (not drawn to scale) below the top DHS profile and show a super-enhancer, a gene, an intragenic CTCF-binding site and an intergenic CTCF-binding site at the indicated resolution. The ovals in the left matrices highlight interactions between the constitutive elements of the super-enhancer, which are significantly reduced upon Mediator depletion (top: *P* = 0.00809; bottom: *P* = 0.01393; two-sided unpaired *t*-test). The arrowheads in the middle two matrices highlight the appearance of CTCF-mediated insulation stripes within the gene body following loss of Mediator. The squares in the right matrices highlight regular nucleosome interactions surrounding an intergenic CTCF-binding site, which do not change after Mediator depletion.

## Comparison of Mediator loss and transcription inhibition

The reduction in enhancer-promoter interactions that we observe after Mediator depletion is associated with a strong decrease in gene expression. A plausible explanation for these observations is that weakening of enhancer-promoter interactions leads to lower levels of gene activity. However, it is also possible that reduced transcriptional activity leads to weakening of enhancer-promoter interactions. To get more insight into the cause–consequence relationship between regulatory interactions and transcription, we performed MCC experiments in cells treated with triptolide, which inhibits initiation of transcription (Extended Data Fig. 9). Comparison of the MCC data from DMSO-treated cells with those from triptolide-treated cells shows that chemical inhibition of transcription does not lead to a reduction of enhancer-promoter interactions. By contrast, we find that enhancer-promoter interactions are significantly weaker in cells in which Mediator is depleted than in cells in which transcription is inhibited. This indicates that enhancer-promoter interactions are dependent on Mediator and not on the process of transcription.

Although chemical inhibition of transcription does not result in reduced enhancer-promoter interactions, we observe increased interactions with CTCF-binding sites following triptolide treatment. This indicates that it is possible that the increased CTCF-mediated interactions, which we detect after Mediator depletion, result from reduced transcription in the locus.

## BET proteins do not compensate for depletion of Mediator

Our data show that both short- and long-range interactions between enhancers and promoters are dependent on Mediator. However, we find that enhancer-promoter interactions are not completely abolished when Mediator is depleted. This indicates that other factors are involved in mediating enhancer-promoter interactions and possibly compensate for the loss of Mediator. It has recently been suggested that BRD4 plays a role in genome organization and stabilizes Cohesin on chromatin[42]. Although it has been shown that inhibition of BET proteins alone does not lead to changes in enhancer-promoter interactions (despite having a strong impact on transcription)[43], we wondered

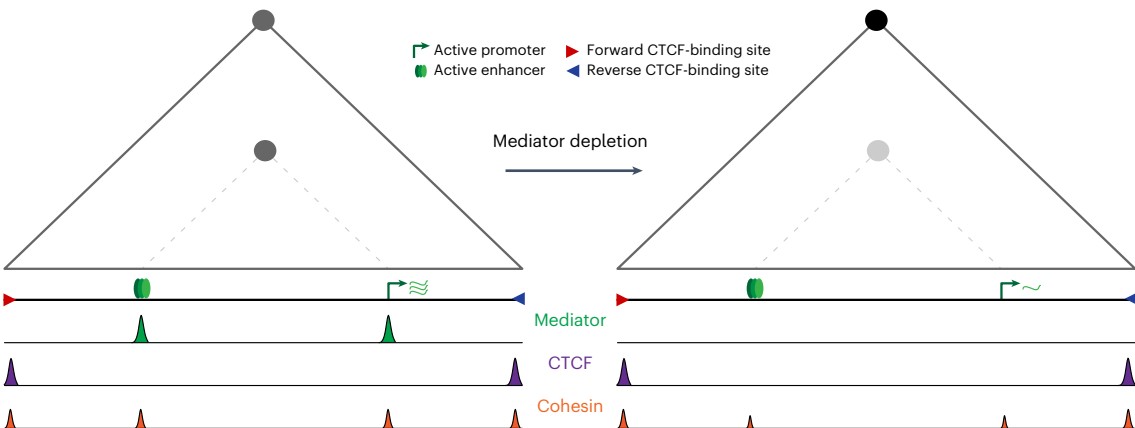

**Fig. 6 | Graphical summary.** The panels show a schematic TAD (gray triangle), interactions between the CTCF-binding sites located at its boundaries (gray circle at the TAD apex) and enhancer-promoter interactions (gray circle at the intersection between the enhancer and promoter, as indicated with a dashed line). Upon Mediator depletion, Cohesin occupancy at the enhancer and promoter is reduced, and enhancer-promoter interactions are weakened. By contrast, the TAD structure remains intact and the interactions between the CTCF-binding sites at the TAD boundaries are increased.

whether Mediator and BET proteins might have (partly) redundant roles in enhancer-promoter interactions. We therefore investigated the impact of combined Mediator depletion and chemical BET inhibition on enhancer-promoter interactions with Capture-C (Extended Data Fig. 10). However, we do not find consistent additional effects on enhancer-promoter interactions after combined Mediator depletion and BET inhibition, compared with depletion of Mediator alone. This suggests that enhancer-promoter interactions result from a more complex interplay between many regulatory factors.

## Discussion

In this study, we have investigated the function of the Mediator complex in the regulation of chromatin architecture and enhancer-promoter interactions (Fig. 6). To overcome limitations of existing studies[24–29], we have combined rapid depletion of Mediator using dTAG technology and analysis of genome architecture at very high resolution with targeted MNase-based 3C approaches. This strategy has enabled us to demonstrate that depletion of Mediator leads to a significant reduction in enhancer-promoter interactions.

We have focused our analyses on 20 gene loci containing strong super-enhancers and found an average decrease in interaction strength of ~34% between promoters and Mediator-bound enhancer elements in these regions. This reduction in enhancer-promoter interactions is associated with an average downregulation of expression of ~7.5-fold for the genes we investigated. The relatively small effect on interaction frequency in comparison with gene activity is in agreement with recent studies that have shown that the relationship between enhancer-promoter interaction frequency and transcriptional output is not linear and that small changes in genome architecture can have a large impact on gene activity levels[44,45].

In the context of Mediator depletion, there are several possible explanations for these observations. We have focused our analyses on genes regulated by super-enhancers, which are composed of many individual elements. For example, the *MTAP* gene is regulated by two super-enhancers, which contain more than twenty individual active elements. The additive and potentially synergistic impact of reduced interactions of each of these elements could cumulatively cause large changes in gene expression levels. In addition, the Mediator complex plays a central role in the regulation of gene expression and is thought to act at several stages of the transcription cycle. It is therefore likely that the large decrease in transcriptional output upon Mediator depletion is related not only to weaker enhancer-promoter interactions, but also to the loss of the general function of Mediator in the regulation of initiation (for example, PIC assembly and activation), re-initiation, elongation and transcriptional bursting[23,46]. Moreover, it is thought that the function of the Mediator complex in gene regulation is (partly) dependent on the formation of nuclear condensates[47–51]. In agreement with this model, it has been shown that MED14 depletion leads to dissolved Pol II clusters[28]. It is possible that the reduced interactions between enhancers and promoters after Mediator loss are not sufficient to establish the required concentrations of transcription factors, coactivators and Pol II for the formation of nuclear condensates in which transcription can be efficiently initiated. Finally, it is important to note that enhancer-promoter interactions are thought to be transient and vary from cell to cell[2]. It has been shown that enhancer-promoter proximity does not necessarily co-occur with transcriptional burst[52]; the precise mechanisms by which interactions between enhancers and their target gene promoter relate to transcriptional activation therefore require further investigation.

Our data indicate that Mediator's role in enhancer-promoter interactions is (partly) dependent on Cohesin. Although it has previously been shown that Mediator co-localizes with Cohesin[24,25], the functional relationship between these complexes has thus far been unclear. Our data show that Cohesin levels at enhancers are reduced when Mediator is depleted. A possible explanation for this observation is that Mediator stabilizes Cohesin on chromatin. Although further investigation of the interaction between Mediator and Cohesin is required, this suggests that Cohesin and Mediator cooperate in the formation of enhancer-promoter interactions and provides support for a model in which extruding Cohesin molecules are stalled at Mediator-bound enhancers and promoters and thereby bridge interactions between these elements. These findings indicate that Cohesin extrusion trajectories are dependent on multiple regulatory proteins and that these factors cooperate in the formation of specific 3D chromatin structures in which gene expression is regulated[53].

The high resolution of our data has enabled us to visualize the effects of Mediator depletion on nano-scale genome organization. We find that interactions between the individual elements within super-enhancers and interactions between enhancers and promoters across very small distances are dependent on Mediator. Of note, we have previously shown that Cohesin depletion leads to a reduction of enhancer-promoter interactions across medium and large genomic distances (>~10 kb), but that Cohesin is not involved in regulating short-range enhancer-promoter interactions or interactions within enhancer clusters[14]. This suggests that Cohesin has a role in facilitating longer-range enhancer-promoter interactions and that Mediator can function independently on smaller scales. At the level

of nano-scale genome organization, we also detect changes in chromatin structure at CTCF-binding sites. Since we only observe these changes at CTCF-binding sites within gene bodies and not at intergenic CTCF-binding sites, we think that these changes are related to the reduced transcription levels following Mediator depletion. This implies that specific higher-order nucleosome structures within genes can form only in the absence of high transcriptional activity, which is consistent with experiments in yeast that have shown that the transcriptional machinery disrupts regular nucleosome spacing[54].

Although the changes in chromatin structure at intragenic CTCF-binding sites are likely related to lower transcription levels that result from Mediator depletion, it is important to note that we do not observe a reduction in enhancer-promoter interactions following treatment with triptolide to chemically inhibit transcription. These observations are consistent with several recent reports in which the impact of acute inhibition of transcription was analyzed with high-resolution Micro-C approaches[55,56]. This indicates that enhancer-promoter interactions depend on the Mediator complex and not on the process of transcription. However, it is of interest that we observe increased interactions with CTCF-binding sites following transcription inhibition. This suggests that the increased CTCF-mediated interactions that we detect after Mediator depletion could be related to the reduced levels of transcription that are associated with loss of Mediator. A possible explanation for these observations is that transcribing Pol II molecules form an obstacle to extruding Cohesin molecules; CTCF loops might therefore form more efficiently when transcription levels are reduced. This model fits with previous work that has shown that the distribution of Cohesin is dependent on transcription[57,58] and with two recent reports indicating that Pol II can form barriers to loop extrusion[59,60].

With the exception of a subtle increase in the strength of TAD and sub-TAD boundaries, we do not observe large-scale changes in genome architecture upon Mediator depletion. This is consistent with previous reports, in which the impact of Mediator depletion has been investigated with lower resolution approaches, such as Hi-C and Hi-ChIP[27–29]. On the basis of knockout of the Mediator-CDK module, it has recently been suggested that the Mediator complex is involved in the regulation of heterochromatin domains and genome compartmentalization[61]. We do not observe clear changes in compartmentalization after 2 h of Mediator depletion, but it is likely that changes in compartmentalization would require more time to manifest[62–64].

Although our data clearly show that enhancer-promoter interactions are dependent on Mediator, we do not observe a complete loss of interactions when Mediator is depleted. This suggests that additional proteins and mechanisms play a role in mediating enhancer-promoter interactions. We find that the interactions that remain following depletion of Mediator are not dependent on BET proteins. However, many other regulatory factors, such as tissue-specific transcription factors[65–67] and more widely expressed transcription factors, such as LDB1 (refs. [68–71]) and YY1 (refs. [72,73]), have been implicated in enhancer-promoter interactions. It is likely that the regulation of enhancer-promoter interactions is dependent on a complex interplay between multiple regulatory proteins, which might act in a (partly) redundant manner to ensure the formation of robust enhancer-promoter interactions. In line with biochemical and structural evidence[20,21], our data show that the Mediator complex is one of the factors with an important role in regulating enhancer-promoter communication and gene expression, by acting as both a functional and an architectural bridge between enhancers and promoters.

## Online content

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

## Methods

### Cell culture

Wild-type and MED14-dTAG human colorectal carcinoma HCT-116 cells[28] were cultured in RPMI 1640 medium (Gibco, 21875034) supplemented with 10% FBS (Gibco, 10270106) and 1× penicillin–streptomycin (Gibco, 15140122) at 37 °C and 5% $CO_2$. Cells were passaged every 2–3 d by trypsinization (Gibco, 25300054) upon reaching ~70–80% confluency. For MED14 depletion, dTAG stock was prepared by dissolving the dTAG$^v$-1 ligand (Tocris, 6914) in DMSO. HCT-116 MED14-dTAG cells were seeded in culture flasks and grown to ~70% confluency. On the day of depletion, the cells were washed once with PBS, replenished with fresh culture medium containing either DMSO only or dTAG ligand at a final concentration of 0.5 μM, and treated for 2 h. For transcription inhibition, triptolide (Sigma, T3652) stock was prepared by dissolving the drug in DMSO, and HCT-116 MED14-dTAG cells were treated with a final concentration of 1 μM for 45 min, as described previously[56]. For co-inhibition of BET proteins, treatment of HCT-116 MED14-dTAG cells with dTAG ligand, as described above, was combined with I-BET 151 dihydrochloride (Tocris, 4650) treatment at 1 μM final concentration for 90 min.

### Immunoblotting

To confirm efficient Mediator depletion, we performed immunoblotting experiments of whole-cell lysates of HCT-116 MED14-dTAG cells treated with dTAG ligand for 0.5, 1, 2, 4, 6 or 8 h. Following treatment, the cells were trypsinized and pelleted. The cell pellets were washed once with PBS and lysed in radioimmunoprecipitation assay (RIPA) lysis and extraction buffer (Thermo Scientific, 89900) supplemented with 250 U mL$^{-1}$ benzonase (Sigma-Aldrich, E1014) and protease inhibitor cocktail containing leupeptin (Carl Roth, CN33.4), PMSF (Carl Roth, 6367.3), pepstatin A (Carl Roth, 2936.3) and benzamide hydrochloride (Acros Organics, E1014) for 1 h at 4 °C on a rotator. Lysates were cleared by centrifugation at maximum speed for 15 min at 4 °C. Protein concentration was measured using Bio-Rad protein assay kit (Bio-Rad, 5000006). For each sample, 20 μg of protein lysate was mixed with 4X LDS sample buffer (Invitrogen, NP0007) supplemented with 50 mM DTT (Carl Roth, 6908.3) and denatured for 5 min at 95 °C. Proteins were separated on a NuPAGE 4–12% Bis-Tris gel (Invitrogen, NP0321) and blotted to a PVDF membrane. The membrane was blocked with 5% milk (Carl Roth, T145.2) in 1× PBS containing 0.05% Tween-20 (PBST) for 1 h at room temperature and was cut into two parts to detect higher- and lower-molecular-weight target proteins separately. Cut membranes were incubated with primary antibodies (MED14-HA: 1:1,000, rabbit anti-HA-Tag (C29F4) antibody, Cell Signaling Technology, 3724; GAPDH: 1:2,000, mouse anti-GAPDH antibody (6C5), Abcam, ab8245) at 4 °C overnight. The next day, the membranes were washed three times with PBST and incubated with horseradish peroxidase (HRP)-labeled secondary antibodies (MED14-HA: 1:3,000, goat anti-rabbit IgG H&L (HRP), Abcam, ab205718; GAPDH: 1:3,000, goat anti-mouse IgG H&L (HRP), Abcam, ab205719) for 1 h at room temperature. The membranes were washed three times with PBST again and were developed and imaged using INTAS ChemoCam Imager HR.

To further evaluate the efficiency of Mediator depletion, we performed immunoblotting experiments of subcellular fractions (chromatin, nucleoplasm and cytoplasm) of HCT-116 MED14-dTAG cells treated with dTAG ligand for 2 h[74]. After treatment, the cells were trypsinized and pelleted. Cell pellets were resuspended in cell lysis buffer (10 mM Tris-HCl pH 7.4, 150 mM NaCl, 0.15% NP-40, 1× protease inhibitor mix) and incubated on ice for 5 min. The resulting cell lysates were gently transferred to fresh protein LoBind tubes containing 2.5 volumes of cold sucrose buffer (10 mM Tris-HCl pH 7.4, 150 mM NaCl, 24% sucrose, 1× protease inhibitor mix). After centrifugation, the supernatants were collected and stored as cytoplasmic fractions. The resulting nuclei pellets were resuspended in glycerol buffer (20 mM Tris-HCl pH 7.4, 75 mM NaCl, 0.5 mM EDTA, 50% glycerol, 1× protease inhibitor mix),

to which nuclear lysis buffer (10 mM Tris-HCl pH 7.4, 300 mM NaCl, 0.2 mM EDTA, 1 M urea, 7.5 mM MgCl$_2$, 1% NP-40, 1× protease inhibitor mix) was added. After incubation on ice for 2 min, the lysates were centrifuged to precipitate the chromatin–RNA complex. The supernatants were collected and stored as nucleoplasmic fractions. The resulting chromatin pellets were briefly washed once with MNase buffer (10 mM Tris-HCl pH 7.5, 10 mM CaCl$_2$) and resuspended in pre-warmed chromatin digest buffer (1× MNase buffer, 1× BSA, 50 U μL$^{-1}$ MNase, 100 mM NaCl), followed by incubation at 37 °C and 1,400 r.p.m. for 3 min. The digestion reactions were quenched by the addition of 25 mM EGTA and centrifuged, and the supernatants were collected and stored as chromatin fractions. The fractions were analyzed by immunoblotting, as described above. The following primary antibodies were used: MED-HA: 1:1,000, rabbit anti-HA-Tag (C29F4) antibody (Cell Signaling Technology, 3724); GAPDH: 1:2,000, mouse anti-GAPDH (6C5) antibody (Abcam, ab8245); and histone H3: 1:5,000, rabbit HRP anti-histone H3 antibody (Abcam, ab21054). All immunoblotting experiments were performed independently for at least three times, with similar results.

### ChIP–seq

Calibrated MNase ChIP–seq was performed as described previously[75], with some modifications for three biological replicates per experimental condition. Fresh protease (Roche, 11873580001) and phosphatase inhibitors (Roche, 4906837001) were added to all buffers. Briefly, $6 × 10^7$ cells were crosslinked with 1% formaldehyde for 8 min at room temperature, followed by quenching with 125 mM glycine for 5 min. The fixed cells were scraped from the plates, washed twice with ice-cold PBS and centrifuged. The cell pellets were resuspended in Farnham lysis buffer (5 mM PIPES pH 8, 85 mM KCl, 0.5% NP-40) and incubated on ice for 10 min. After centrifugation, the nuclei pellets were resuspended in 1% SDS lysis buffer (50 mM Tris-HCl pH 8, 10 mM EDTA, 1% SDS). Following incubation at room temperature for 10 min, IP buffer (20 mM Tris-HCl pH 8, 1 mM EDTA, 150 mM NaCl, 1% Triton X-100) supplemented with 5 mM CaCl$_2$ was added to quench the reaction and to further dilute the SDS (0.1% final concentration). The samples were then digested with 20,000 U of MNase (NEB, M0247S) at 37 °C for 20 min, followed by the addition of 20 mM EDTA and 10 mM EGTA to quench the MNase digestion. The digested samples were sonicated, and the chromatin supernatants were collected afterwards. For each IP, 45 μg of sample chromatin and 200 ng of *Drosophila* S2 MNase-digested chromatin were used. The samples were pre-cleared with Dynabeads Protein G (Thermo Fisher Scientific, 10009D) for 30 min at 4 °C. Pre-cleared samples were incubated with 1.32 μg of rabbit anti-HA-Tag (C29F4) antibody (Cell Signaling Technology, 3724) and 1 μg of *Drosophila* spike-in antibody (Active Motif, 61686) and incubated overnight with gentle rotation. Following incubation, inputs were collected and stored for each sample. The samples were further incubated with Dynabeads Protein G at 4 °C for 3 h. Bead washes were performed at 4 °C for 5 min in the following order: 1× with Buffer 1 (20 mM Tris-HCl pH 8, 2 mM EDTA, 150 mM NaCl, 1% Triton X-100, 0.1% SDS), 4× with Buffer 2 (20 mM Tris-HCl pH 8, 2 mM EDTA, 500 mM NaCl, 1% Triton X-100, 0.1% SDS), 1× with Buffer 3 (10 mM Tris-HCl pH 8, 1 mM EDTA, 250 mM LiCl, 1% NP-40, 1% sodium-deoxycholate), and 3× with TE buffer (10 mM Tris-HCl pH 8, 1 mM EDTA, 50 mM NaCl). The beads were subsequently eluted in elution buffer (0.1 M NaHCO$_3$, 160 mM NaCl, 1% SDS). The samples were de-crosslinked, and DNA extraction was performed. Library preparations were performed using the NEBNext Ultra II DNA Library Prep Kit for Illumina (NEB, E7645S) with a modified thermocycler program for the End Prep reaction (20 °C for 30 min, 50 °C for 1 h; heated lit set to 60 °C). The amplified libraries were size selected with double-sided (1.0-1.2x) SPRI bead purification. The final libraries were assessed on a fragment analyzer and sequenced using the NextSeq550 Illumina platform (43-bp paired-end reads). Paired-end reads were processed for adapter removal and mapped to the hg38 reference genome using Bowtie2[76]. Duplicates were filtered and removed using SAMtools[77].

Spike-ins from *Drosophila* chromatin were used for normalization. Normalized bigwig files were generated using Deeptools[78]. Peak calling was performed with MACS2 (ref. [79]) in DMSO samples using input files for thresholding. Box plots were generated with R using default settings.

## Capture-C

Capture-C was performed as described previously[80,81] for three biological replicates per experimental condition. Briefly, $10 \times 10^6$ cells per biological replicate were crosslinked, followed by cell lysis. 3 C libraries were generated by DpnII digestion and subsequent proximity ligation. After decrosslinking and DNA extraction, the resulting 3 C libraries were sonicated to a fragment size of ~200 bp and indexed with Illumina sequencing adapters, using Herculase II polymerase (Agilent, 600677) for library amplification. To boost library complexity, indexing was performed in two parallel reactions for each sample. Biotinylated oligonucleotides (70 nt) were designed using a python-based oligo tool[82] (https://oligo.readthedocs.io/en/latest/) and used for enrichment of the libraries in two consecutive rounds of hybridization, biotin-streptavidin bead pulldown (Invitrogen, 65306), bead washes and PCR amplification (KAPA HyperCapture Reagent Kit, Roche, 09075828001). The final libraries were assessed on a fragment analyzer and sequenced using the NextSeq550 Illumina platform (75-bp paired-end reads). Data analysis was performed using the CapCruncher pipeline[80] (https://github.com/sims-lab/CapCruncher).

## Micro-Capture-C

Micro-Capture-C (MCC) was performed as described previously[33] for three biological replicates per experimental condition. Briefly, multiple aliquots of $10 \times 10^6$ cells per biological replicate were crosslinked and permeabilized with 0.005% digitonin (Sigma-Aldrich, D141). For each replicate, the permeabilized cells were pelleted, resuspended in nuclease-free water, and split into three digestion reactions. MCC libraries were generated by digesting the chromatin in low $Ca^{2+}$ MNase buffer (10 mM Tris-HCl pH 7.5, 10 mM $CaCl_2$) for 1 h at 37 °C with MNase (NEB, M0247) added in varied concentrations (17–32 Kunitz U). The reactions were quenched by the addition of 5 mM ethylene glycol-bis (2-aminoethylether)-N,N,N′,N′-tetraacetic acid (EGTA) (Sigma-Aldrich, E3889) and pelleted afterwards. The pellets were resuspended in PBS containing 5 mM EGTA, and an aliquot of 200 mL per reaction was tested for digestion efficiency as a control. The reactions were pelleted again and resuspended in DNA ligase buffer (Thermo Scientific, B69) supplemented with dNTP mix (NEB, N0447) at 0.4 mM final concentration and 2.5 mM EGTA. Subsequently, 200 U mL$^{-1}$ T4 polynucleotide Kinase (NEB, M0201), 100 U mL$^{-1}$ DNA polymerase I large (Klenow) fragment (NEB, M0210) and 300 U mL$^{-1}$ T4 DNA ligase (Thermo Scientific, EL0013) were added. The reactions were incubated at 37 °C and 20 °C for 1–2 h and overnight, respectively. Following chromatin decrosslinking, DNA extraction was performed using DNeasy blood and tissue kit (Qiagen, 69504). The size-selected MCC libraries were sonicated, indexed and enriched with a double-capture procedure, as described in 'Capture-C.' Biotinylated oligonucleotides (120 nucleotides) were designed using a python-based oligonucleotide tool[82] (https://oligo.readthedocs.io/en/latest/). The final libraries were assessed on a fragment analyzer and were sequenced using the NextSeq550 Illumina platform (150-bp paired-end reads). Data analysis was performed using the MCC pipeline[33].

## Tiled Micro-Capture-C

Tiled-MCC was performed using the generated MCC libraries, following a tiled enrichment procedure as described previously[14], using the Twist Hybridization and Wash Kit (Twist Bioscience, 101025). Briefly, indexed MCC libraries were pooled and dried completely in a vacuum concentrator at 45 °C. Dried DNA was resuspended in blocker solution and pooled with the hybridization solution containing a custom panel of biotinylated oligonucleotides (70 nt; designed using a python-based oligo tool[82] (https://oligo.readthedocs.io/en/latest/) and incubated at 70 °C overnight. Streptavidin bead pulldown and bead washes were performed with Twist Wash Buffers according to the manufacturer's instructions (Twist Target Enrichment Protocol). Subsequently, post-hybridization PCR was performed with 11 cycles of amplification. PCR-amplified libraries were purified using pre-equilibrated Twist DNA Purification Beads. The final libraries were assessed on a fragment analyzer and sequenced using the NextSeq550 Illumina platform (150-bp paired-end reads). Data analysis was performed using the MCC pipeline[33] (https://github.com/jojdavies/Micro-Capture-C) and HiC-Pro pipeline[83] (https://github.com/nservant/HiC-Pro) as described previously[14]. All contact matrices were balanced using ICE-normalization[84]. The large-scale contact matrices have a resolution of 500 bp – 2 kb (depending on the size of the region); the resolution of the nano-scale matrices is indicated in the figures.

## ATAC-seq

Assay for Transposase-Accessible Chromatin using sequencing (ATAC-seq) was performed as described previously[34,85] with some modifications. Three biological replicates per experimental condition were used for the experiment. Briefly, $1.5 \times 10^5$ washed cells were split over two tubes, followed by centrifugation. Cell pellets were resuspended in fresh cold lysis buffer (10 mM Tris-HCl pH 7.5, 10 mM NaCl, 3 mM $MgCl_2$, 0.1% Igepal CA-630) and incubated on ice for 3 min. The lysates were washed once with cold PBS, and the resulting nuclear pellets were resuspended in the tagmentation mix (Illumina, 20034198). The tagmentation reactions were performed at 37 °C and 1,000 r.p.m. for 30 min, followed by DNA purification using MinElute PCR purification kit (Qiagen, 28004). The indexed samples were amplified using Nextera indexing primers and NEBNext High-Fidelity PCR Master Mix (NEB, M0541), with an initial 5-min extension step at 72 °C. A real-time PCR library amplification kit (KAPA, KK2701) was used to calculate the required number of PCR cycles (11 cycles) in order to minimize library amplification bias. Size selection was performed with double-sided SPRI bead purification to remove primer dimers and larger fragments (>700 bp). The final libraries were assessed on a fragment analyzer and sequenced using the NextSeq550 Illumina platform (75-bp paired-end reads). The data from each replicate were down-sampled to the library with the lowest read depth and analyzed using the NGseqBasic pipeline[86].

## CUT&Tag

CUT&Tag[40] was performed for three biological replicates (for a total of five technical replicates) per experimental condition using the CUT&Tag-IT Assay Kit (Anti-Rabbit) (Active Motif, 53160), according to the manufacturer's instructions with some modifications. Briefly, $0.5 \times 10^6$ cells were mildly crosslinked with 0.3% paraformaldehyde (Science Services, E15710), followed by quenching with 125 mM cold glycine. Meanwhile, concanavalin A beads were prepared, following the manufacturer's instructions. The fixed cells were washed, resuspended in wash buffer and incubated with concanavalin A beads for 10 min on a rotator at room temperature. The samples were placed on a magnetic stand to clear the liquid, and the samples were resuspended with ice-cold antibody buffer supplemented with protease inhibitor cocktail and digitonin. Then, 1 μg rabbit anti-SMC1A antibody (1:50, Abcam, ab9262) or 1 μg rabbit IgG isotype control antibody (1:50, Cell Signaling Technology, 2729S) was added to each sample, and the samples were incubated overnight at 4 °C on a rotator in 0.2-mL PCR tubes. The next day, the samples were incubated with guinea pig anti-rabbit secondary antibody (1:100, Active Motif, 53160) for 1 h at room temperature on a rotator, followed by washes with dig-wash buffer. The samples were placed on a magnetic stand to clear the liquid, and the beads were resuspended with CUT&Tag-IT Assembled pA-Tn5 Transposons. The reactions were subsequently incubated at room temperature on a rotator, followed by washes with Dig-300 buffer. After clearing the liquid

on a magnetic stand, the beads were resuspended with tagmentation buffer. The tagmentation reactions were subsequently incubated at 37 °C for 60 min. The samples were de-crosslinked, and DNA extraction was performed according to the manufacturer's instructions. The libraries were amplified by PCR, and size selection was performed with two rounds of SPRI bead purification to remove primer dimers. The final libraries were assessed on a fragment analyzer and sequenced using the NextSeq550 Illumina platform (75-bp paired-end reads). The data were analyzed using the NGseqBasic pipeline[86]. Peak calling was performed with MACS2 (ref. 79) using IgG controls for thresholding. Normalized bigwig files and meta peak profiles were generated using Deeptools[78] and LOESS regression was applied for smoothening of the data. Box plots were generated with R using default settings. Differential binding analysis was performed in R using the DiffBind package. An adjusted *P* value of 0.05 (Benjamini–Hochberg method) was used to identify differentially bound SMC1A peaks after Mediator depletion (Supplementary Table 1).

### Public data analysis
DNase-I hypersensitivity data[87] (ENCSR000ENM) and ChIP–Seq data for CTCF[87] (ENCSR000BSE) and MED26 (ref. 27) in HCT-116 cells were analyzed using the NGseqBasic pipeline[86]. TT-seq data files for HCT-116 MED14-dTAG cells[28] were shared by the authors, and differential expression analysis was performed in R using the DESeq2 package[88].

### Reporting summary
Further information on research design is available in the Nature Portfolio Reporting Summary linked to this article.

### Data availability
The raw sequencing and processed data are available from the Gene Expression Omnibus (GEO) as a SuperSeries under accession number GSE205984. DNase-I hypersensitivity data[87] and ChIP–seq data for CTCF[87] are available from ENCODE under accession codes ENCSR000ENM and ENCSR000BSE, respectively. ChIP–seq data for MED26 (ref. 27) are available from GEO under accession code GSE121355. TT-seq data[28] are available from GEO under accession code GSE139468. Source data are provided with this paper.

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

### Acknowledgements
We would like to thank M. Jäger and G. Winter (CeMM Research Center for Molecular Medicine of the Austrian Academy of Sciences, Vienna, Austria) for providing the HCT-116 MED14-dTAG cell line. We are grateful to P. Cramer for discussions, advice and infrastructure support. We would like to thank M. Jäger, K. Lysakovskaia, K. Maier, M. Rohm, P. Rus, N. Übelmesser and K. Zumer for experimental advice and support. We are also grateful to J. Dekker, D. Higgs, E. Oberbeckmann, A. Papantonis and J. Söding for helpful discussions and feedback. A.M.O. is supported by the Max Planck Society and the Deutsche Forschungsgemeinschaft (DFG) via SFB 1565 (Projektnummer 469281184; project P02).

### Author contributions
S.R. performed experiments, analyzed data and wrote the manuscript. A.A., M.A.K., T.B.N.C., T.V. and J.N.C. performed experiments. M.L. analyzed data. A.M.O. conceived and supervised the project, analyzed data and wrote the manuscript. All authors edited and contributed to the manuscript.

### Funding

### Competing interests
The authors declare no competing interests.

### Additional information
**Extended data** is available for this paper at https://doi.org/10.1038/s41594-023-01027-2.

**Correspondence and requests for materials** should be addressed to A. Marieke Oudelaar.

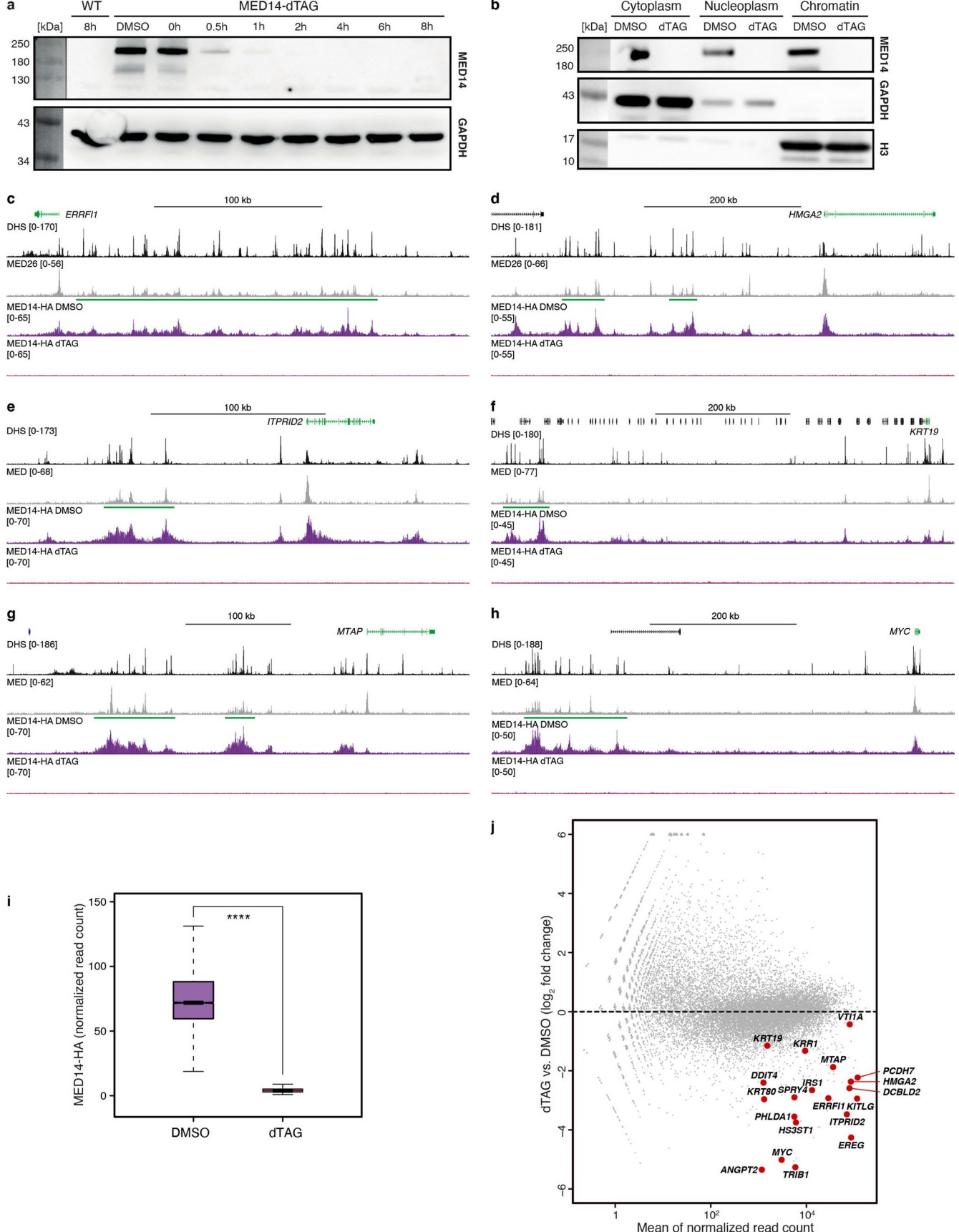

**Extended Data Fig. 1 | See next page for caption.**

**Extended Data Fig. 1 | Characterization of HCT-116 MED14-dTAG cells by immunoblotting, ChIP-seq and TT-seq. a**. A representative immunoblot blot for MED14-dTAG-HA in whole-cell lysates of wildtype (WT) and MED14-dTAG HCT-116 cells treated with DMSO for two hours or dTAG ligand for various durations as specified. MED14-dTAG-HA is detected using an anti-HA primary antibody. GAPDH is shown as loading control. The experiment was repeated independently three times with similar results. **b**. A representative immunoblot blot for MED14-dTAG-HA in subcellular fractions (cytoplasm, nucleoplasm and chromatin) of MED14-dTAG HCT-116 cells treated with DMSO or dTAG ligand for two hours. MED14-dTAG-HA is detected using an anti-HA primary antibody. GAPDH and histone H3 are shown as loading controls. The experiment was repeated independently three times with similar results. **c**. ChIP-seq data for MED14-HA in the *ERRFI1* locus in HCT-116 MED14-dTAG cells treated with DMSO (dark purple; n = 3 biologically independent samples) or dTAG ligand (light purple; n = 3 biologically independent samples). Gene annotation, DNase hypersensitive sites (DHS) and ChIP-seq data for MED26 are shown above. Super-enhancers are highlighted in green below the MED26 profiles. The axes of the DHS and MED26 ChIP-seq profiles are scaled to signal; the axes of the MED14-HA ChIP-seq profiles are fixed (ranges indicated in square brackets). Coordinates (hg38): chr1:7,995,001-8,260,000. **d**. Data as described in **c** for the *HMGA2* locus. Coordinates (hg38): chr12:65,370,001-65,980,000. **e**. Data as described in **c** for the *ITPRID2* locus. Coordinates (hg38): chr2:181,690,001-181,990,000. **f.** Data as described in **c** for the *KRT19* locus. Coordinates (hg38): chr17:40,880,001-41,560,000. **g**. Data as described in **c** for the *MTAP* locus. Coordinates (hg38): chr9:21,450,001-21,890,000. **h**. Data as described in **c** for the *MYC* locus. Coordinates (hg38): chr8:127,160,001-127,760,000. **i**. Quantification of MED14-HA ChIP-seq peaks in HCT-116 MED14-dTAG cells treated with DMSO (dark purple; n = 3 biologically independent samples) or dTAG ligand (light purple; n = 3 biologically independent samples). The box plot shows the median and the interquartile range (IQR) of the data and the whiskers indicate the 1.5 IQR values. **** $p < 2.2e\text{-}16$ (two-sided Wilcoxon rank sum test). **j**. Differences in nascent transcript levels as measured by TT-seq in HCT-116 MED14-dTAG cells treated with DMSO or dTAG ligand for two hours[28]. Genes of interest are highlighted in red.

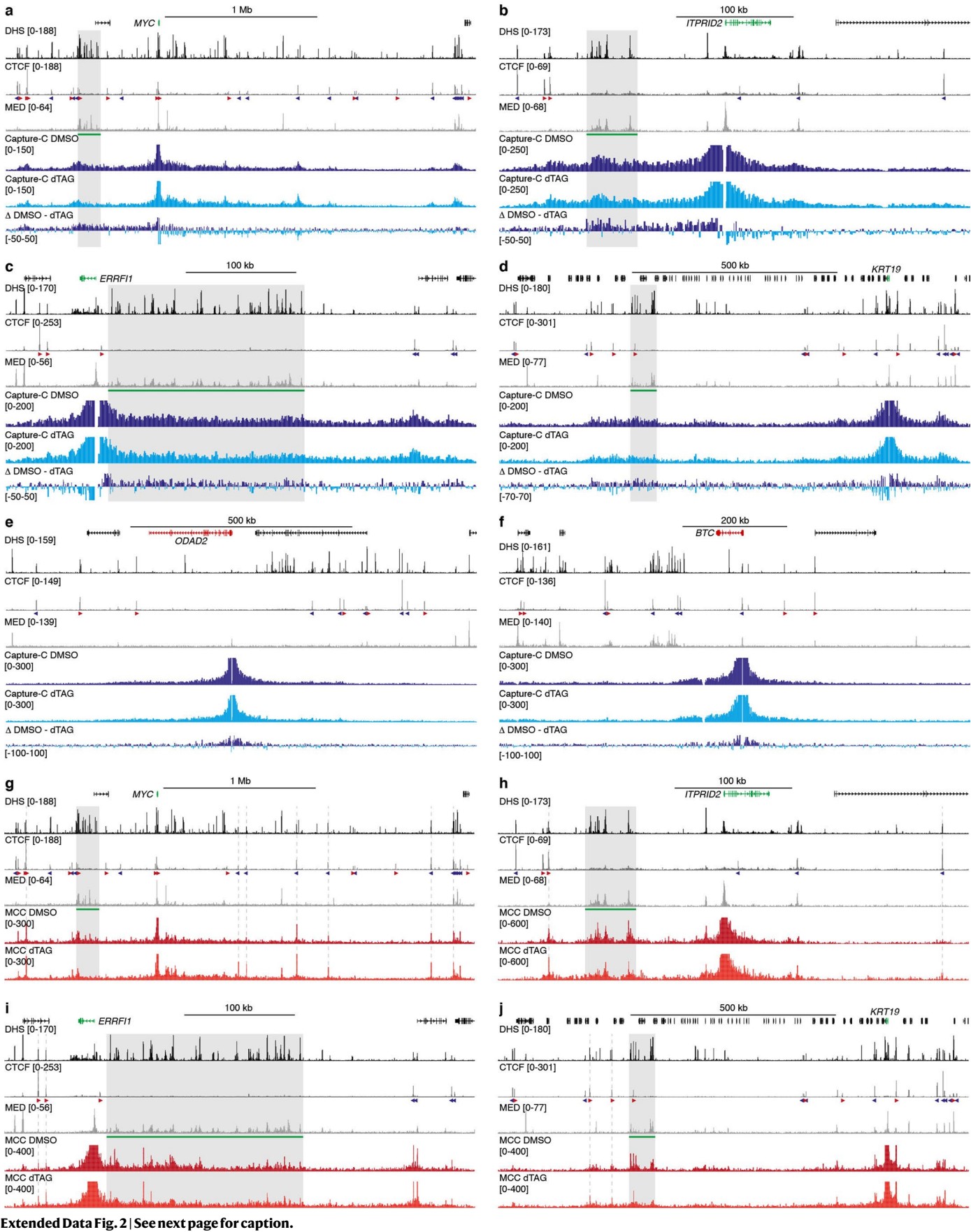

**Extended Data Fig. 2 | See next page for caption.**

**Extended Data Fig. 2 | Capture-C and Micro-Capture-C analysis in HCT-116 MED14-dTAG cells.** Panels **a-d** and **g-j** show regions containing genes that are highly expressed and associated with super-enhancers in HCT-116 cells. The Capture-C and Micro-Capture-C data in these loci show (a tendency for) decreased interactions with the regions containing the super-enhancer. Panels **e-f** show regions containing genes which are not highly expressed in HCT-116 cells and not sensitive to Mediator depletion. The Capture-C data in these regions do not show consistent changes in the interaction patterns (except for some technical variation in the strength of the proximity signals surrounding the promoter viewpoints). **a.** Capture-C interaction profiles from the viewpoint of the *MYC* promoter in HCT-116 MED14-dTAG cells treated with DMSO (dark blue; n = 3 biologically independent samples) or dTAG ligand (light blue; n = 3 biologically independent samples). Annotation as described in Fig. 1. Coordinates (hg38): chr8:126,735,000-129,820,000. **b.** Data as described in **a** for the *ITPRID2* locus. Coordinates (hg38): chr2:181,700,000-182,100,000.

**c.** Data as described in **a** for the *ERRFI1* locus. Coordinates (hg38): chr1:7,945,000-8,370,000. **d.** Data as described in **a** for the *KRT19* locus. Coordinates (hg38): chr17:40,580,000-41,725,000. **e.** Data as described in **a** for the *ODAD2* locus. Coordinates (hg38): chr10:27,490,000-28,550,000. **f.** Data as described in **a** for the *BTC* locus. Coordinates (hg38): chr4:74,330,000-75,230,000. **g.** Micro-Capture-C interaction profiles from the viewpoint of the *MYC* promoter in HCT-116 MED14-dTAG cells treated with DMSO (dark red; n = 3 biologically independent samples) or dTAG ligand (light red; n = 3 biologically independent samples). Annotation as described in Fig. 2. Coordinates (hg38): chr8:126,735,000-129,820,000. **h.** Data as described in **g** for the *ITPRID2* locus. Coordinates (hg38): chr2:181,700,000-182,100,000. **i.** Data as described in **g** for the *ERRFI1* locus. Coordinates (hg38): chr1:7,945,000-8,370,000. **j.** Data as described in **g** for the *KRT19* locus. Coordinates (hg38): chr17:40,580,000-41,725,000.

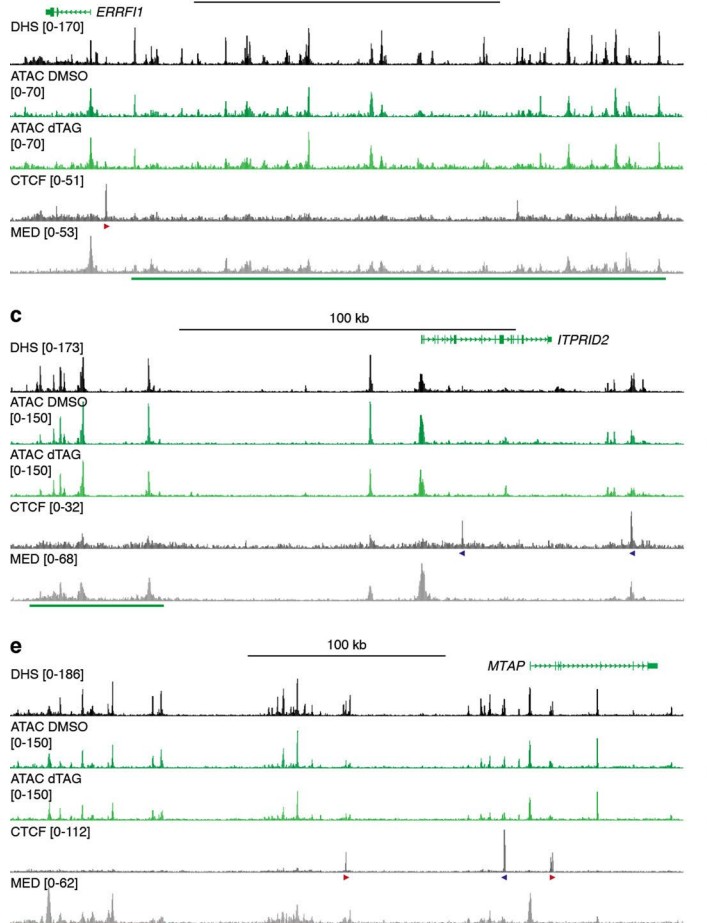

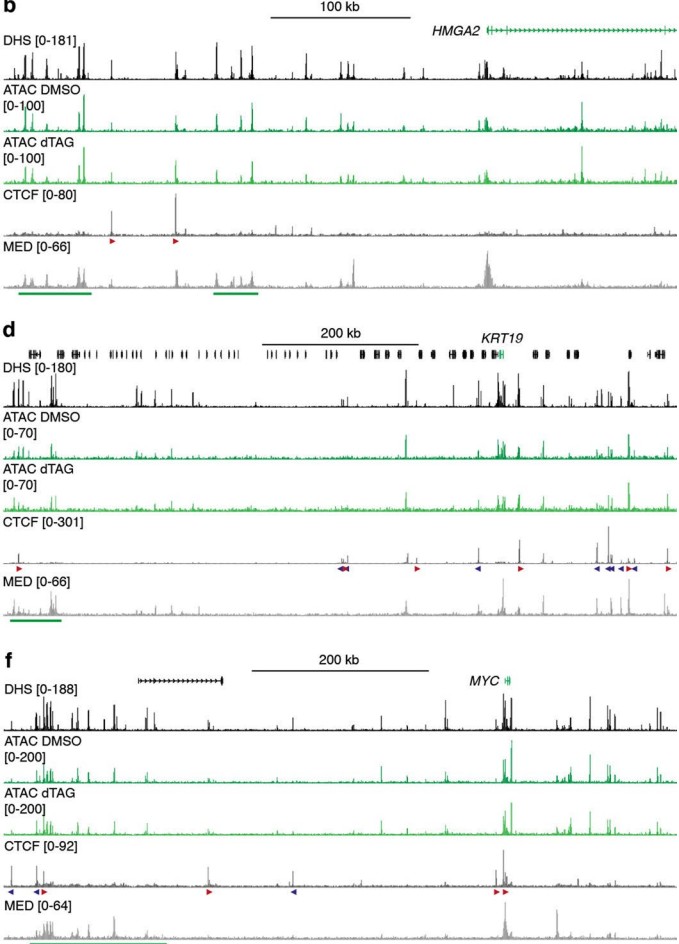

**Extended Data Fig. 3 | ATAC-seq analysis in HCT-116 MED14-dTAG cells.**
**a**. ATAC-seq data in the *ERRFI1* locus in HCT-116 MED14-dTAG cells treated with DMSO (dark green; n = 3 biologically independent samples) or dTAG ligand (light green; n = 3 biologically independent samples). Gene annotation and DNase hypersensitive sites (DHS) are shown above and ChIP-seq data for CTCF and MED26 are shown below. Super-enhancers are highlighted in green below the MED26 profiles and orientations of CTCF motifs are indicated with arrowheads (forward orientation in red; reverse orientation in blue). The axes of the DHS and ChIP-seq profiles are scaled to signal; the axes of the ATAC-seq profiles are fixed (ranges indicated in square brackets). Coordinates (hg38): chr1:8,000,001-8,220,000. **b**. Data as described in **a** for the *HMGA2* locus. Coordinates (hg38): chr12:65,480,001-65,960,000. **c**. Data as described in **a** for the *ITPRID2* locus. Coordinates (hg38): chr2:181,770,001-181,970,000. **d**. Data as described in **a** for the *KRT19* locus. Coordinates (hg38): chr17:40,890,001-41,750,000. **e**. Data as described in **a** for the *MTAP* locus. Coordinates (hg38): chr9:21,540,001-21,880,000. **f**. Data as described in **a** for the *MYC* locus. Coordinates (hg38): chr8:127,170,001-127,930,000.

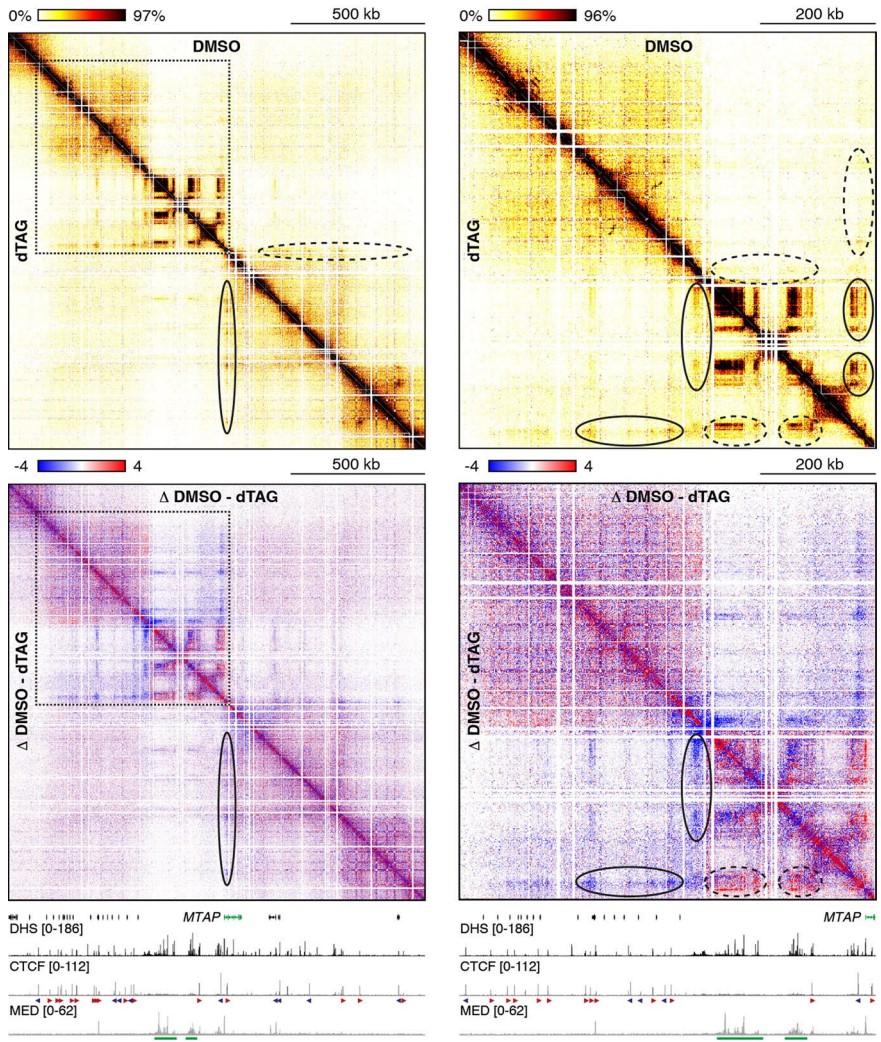

**Extended Data Fig. 4 | Tiled-MCC analysis of the *MTAP* locus in HCT-116 MED14-dTAG cells.** Tiled-MCC contact matrices of the *MTAP* locus in HCT-116 MED14-dTAG cells treated with DMSO (top-right; n = 3 biologically independent samples) or dTAG ligand (bottom-left; n = 3 biologically independent samples). The matrices on the right show a zoomed view of the area indicated by the stippled squares in the left matrices. Differential contact matrices, in which interactions enriched in DMSO-treated cells are shown in red and interactions enriched in dTAG-treated cells are shown in blue, are displayed below. Gene annotation, DNase hypersensitive sites (DHS) and ChIP-seq data for CTCF and MED26 are shown at the bottom. Super-enhancers are highlighted in green below the MED26 profiles and orientations of CTCF motifs are indicated with arrowheads (forward orientation in red; reverse orientation in blue). The dashed black ovals in the dTAG and differential contact matrices highlight decreased enhancer-promoter interactions, whereas the solid ovals indicate increased CTCF interactions. Coordinates (hg38): chr9:21,000,000-22,550,000.

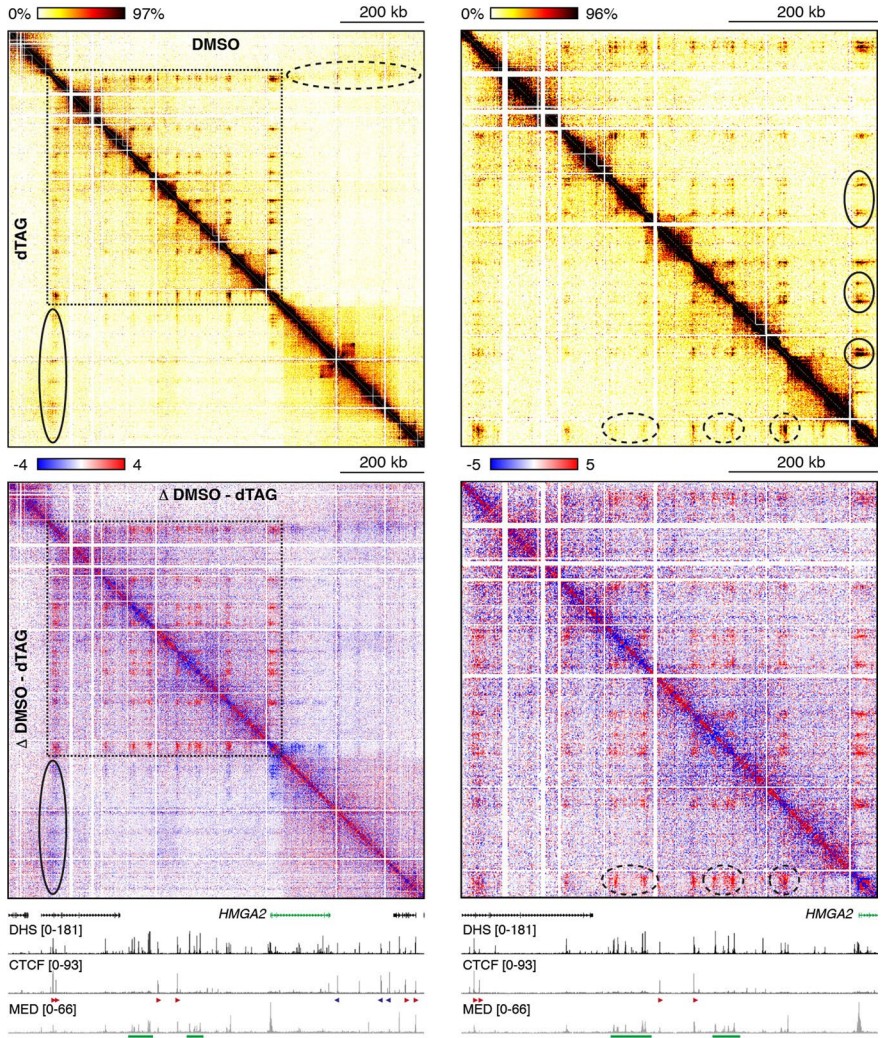

**Extended Data Fig. 5 | Tiled-MCC analysis of the *HMGA2* locus in HCT-116 MED14-dTAG cells.** Tiled-MCC contact matrices of the *HMGA2* locus in HCT-116 MED14-dTAG cells treated with DMSO (top-right; n = 3 biologically independent samples) or dTAG ligand (bottom-left; n = 3 biologically independent samples). The matrices on the right show a zoomed view of the area indicated by the stippled squares in the left matrices. Differential contact matrices, in which interactions enriched in DMSO-treated cells are shown in red and interactions enriched in dTAG-treated cells are shown in blue, are displayed below. Gene annotation, DNase hypersensitive sites (DHS) and ChIP-seq data for CTCF and MED26 are shown at the bottom. Super-enhancers are highlighted in green below the MED26 profiles and orientations of CTCF motifs are indicated with arrowheads (forward orientation in red; reverse orientation in blue). The dashed black ovals in the dTAG and differential contact matrices highlight decreased enhancer-promoter interactions, whereas the solid ovals indicate increased CTCF interactions. Coordinates (hg38): chr12:65,200,000-66,190,000.

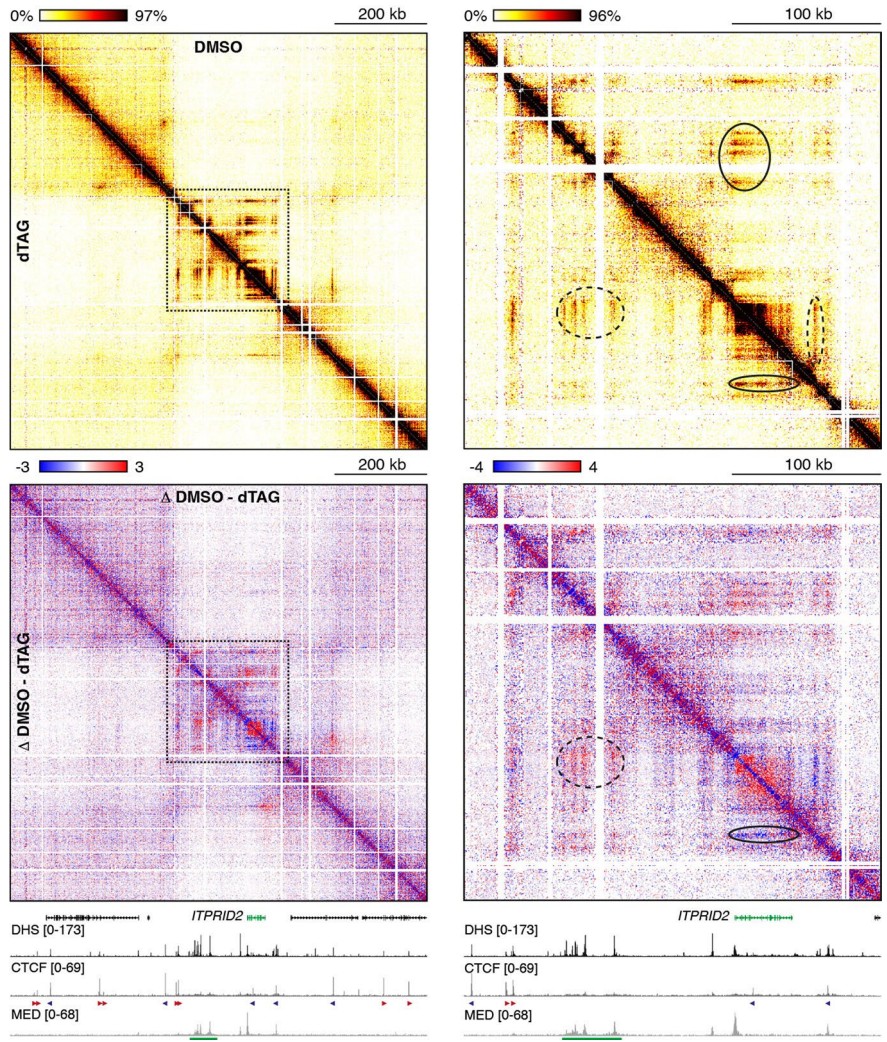

**Extended Data Fig. 6 | Tiled-MCC analysis of the *ITPRID2* locus in HCT-116 MED14-dTAG cells.** Tiled-MCC contact matrices of the *ITPRID2* locus in HCT-116 MED14-dTAG cells treated with DMSO (top-right; n = 3 biologically independent samples) or dTAG ligand (bottom-left; n = 3 biologically independent samples). The matrices on the right show a zoomed view of the area indicated by the stippled squares in the left matrices. Differential contact matrices, in which interactions enriched in DMSO-treated cells are shown in red and interactions enriched in dTAG-treated cells are shown in blue, are displayed below.

Gene annotation, DNase hypersensitive sites (DHS) and ChIP-seq data for CTCF and MED26 are shown at the bottom. Super-enhancers are highlighted in green below the MED26 profiles and orientations of CTCF motifs are indicated with arrowheads (forward orientation in red; reverse orientation in blue). The dashed black ovals in the dTAG and differential contact matrices highlight decreased enhancer-promoter interactions, whereas the solid ovals indicate increased CTCF interactions. Coordinates (hg38): chr2:181,380,000-182,280,000.

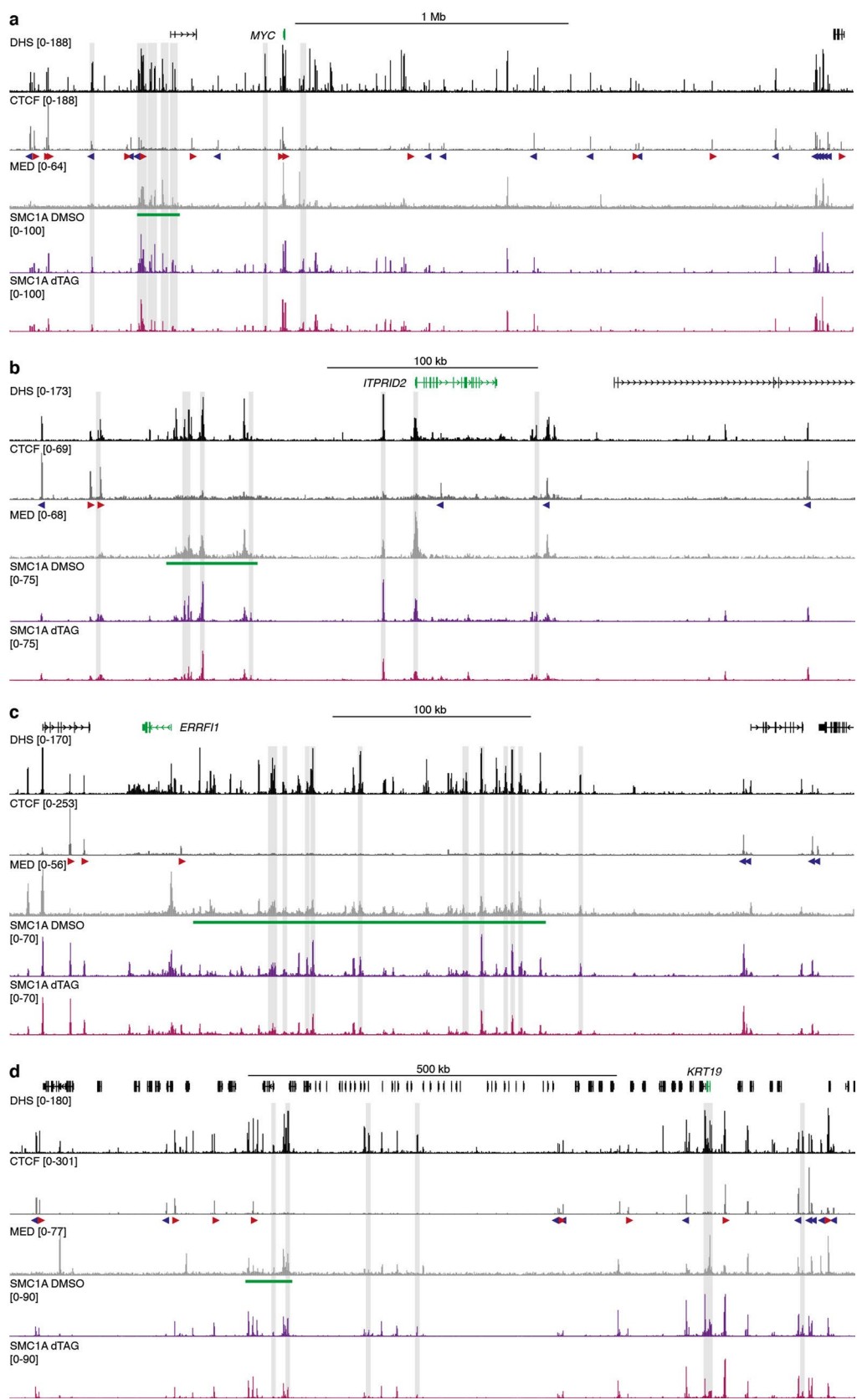

**Extended Data Fig. 7 | See next page for caption.**

**Extended Data Fig. 7 | SMC1A CUT&Tag analysis in HCT-116 MED14-dTAG cells. a**. CUT&Tag data for the Cohesin subunit SMC1A in the *MYC* locus in HCT-116 MED14-dTAG cells treated with DMSO (dark purple; n = 3 biologically independent samples) or dTAG ligand (light purple; n = 3 biologically independent samples). Gene annotation, DNase hypersensitive sites (DHS) and ChIP-seq data for CTCF and MED26 are shown above. Super-enhancers are highlighted in green below the MED26 profiles and orientations of CTCF motifs are indicated with arrowheads (forward orientation in red; reverse orientation in blue). The grey bars highlight SMC1A peaks that are significantly reduced after Mediator depletion (Supplementary Table 1). The axes of the DHS and ChIP-seq profiles are scaled to signal; the axes of the CUT&Tag profiles are fixed (ranges indicated in square brackets). Coordinates (hg38): chr8:126,735,000-129,820,000. **b**. Data as described in **a** for the *ITPRID2* locus. Coordinates (hg38): chr2:181,700,000-182,100,000. **c**. Data as described in **a** for the *ERRFI1* locus. Coordinates (hg38): chr1:7,945,000-8,370,000. **d**. Data as described in **a** for the *KRT19* locus. Coordinates (hg38): chr17:40,580,000-41,725,000.

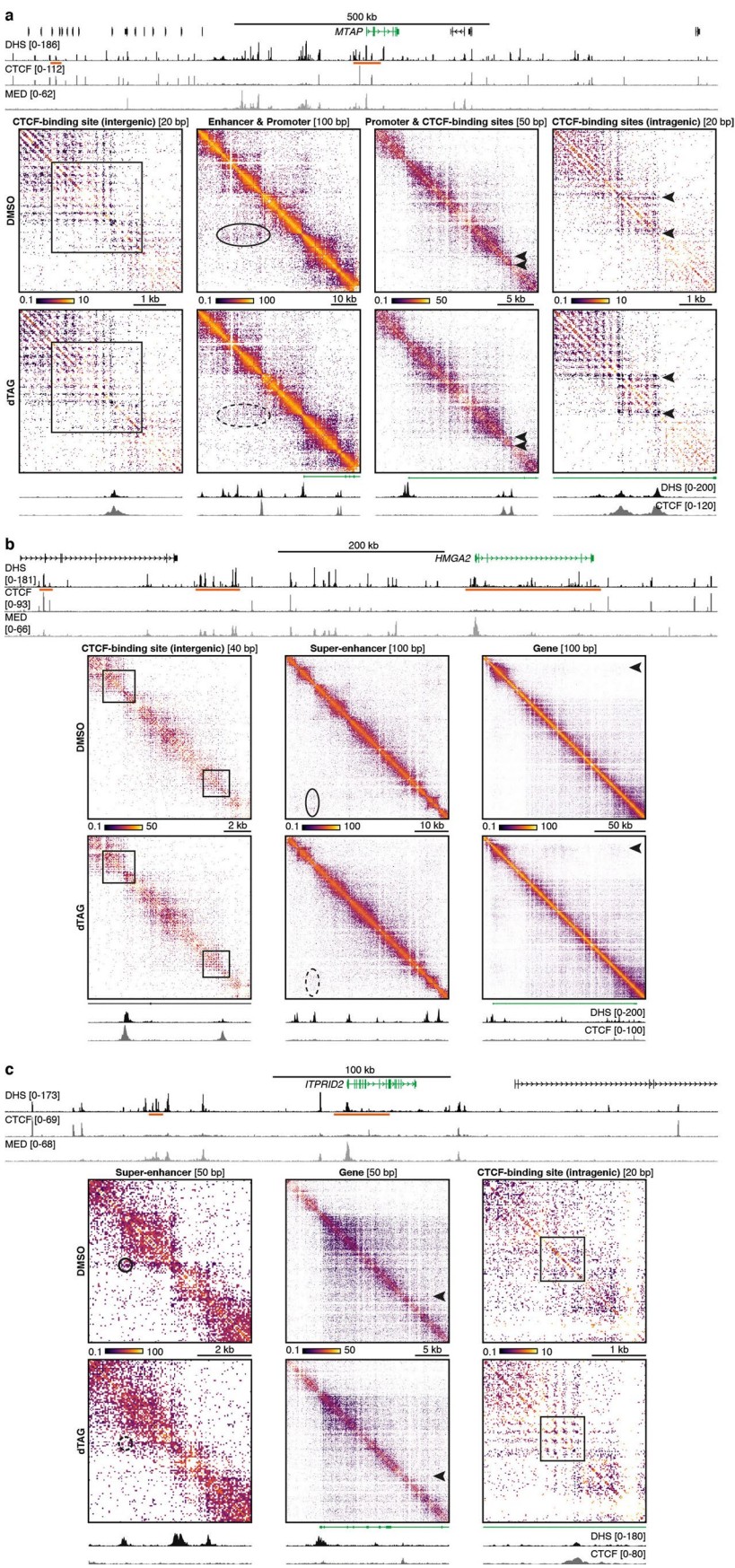

**Extended Data Fig. 8 | See next page for caption.**

**Extended Data Fig. 8 | Micro-topology analysis of the *MTAP*, *HMGA2* and *ITPRID2* loci in HCT-116 MED14-dTAG cells. a**. Tiled-MCC ligation junctions in the *MTAP* locus in HCT-116 MED14-dTAG cells treated with DMSO (top; n = 3 biologically independent samples) or dTAG ligand (bottom; n = 3 biologically independent samples), displayed in localized contact matrices at high resolution. Annotation as described in Fig. 5. The regions covered in the contact matrices show an intergenic CTCF-binding site, an interacting enhancer and promoter, a promoter and CTCF-binding sites and intragenic CTCF-binding sites, at the indicated resolution. The squares in the left matrices highlight regular nucleosome interactions surrounding an intergenic CTCF-binding site, which do not change upon Mediator depletion. The ovals in the middle-left matrices highlight very short-range enhancer-promoter interactions, which are significantly reduced following loss of Mediator ($p = 0.030713$; two-sided unpaired t-test). The arrows in the right two matrices highlight the appearance of CTCF-mediated insulation stripes and CTCF-mediated short-range interactions within the gene body after Mediator depletion. These patterns are difficult to appreciate at the lower resolution displayed in the middle-right matrix, but more clearly visible in the right matrix. **b**. Data as described in **a** for the *HMGA2* locus.

The regions covered in the contact matrices show intergenic CTCF-binding sites, a super-enhancer and a gene, at the indicated resolution. The squares in the left matrices highlight regular nucleosome interactions surrounding intergenic CTCF-binding sites, which do not change upon Mediator depletion. The ovals in the middle matrices highlight interactions between the constitutive elements of the super-enhancer, which are significantly reduced following loss of Mediator ($p = 0.000246$; two-sided unpaired t-test). The arrows in the right two matrices highlight the appearance of specific interaction patterns within the gene body after Mediator depletion. **c**. Data as described in **a** for the *ITPRID2* locus. The regions covered in the contact matrices show a super-enhancer, a gene and an intragenic CTCF-binding site, at the indicated resolution. The circles in the left matrices highlight interactions between the constitutive elements of the super-enhancer, which are significantly reduced upon Mediator depletion ($p = 0.027392$; two-sided unpaired t-test). The arrows and squares in the middle and right matrices highlight the appearance of regular nucleosome patterning surrounding a CTCF-binding site within the gene body following loss of Mediator. These patterns are difficult to appreciate at the lower resolution displayed in the middle matrix, but more clearly visible in the right matrix.

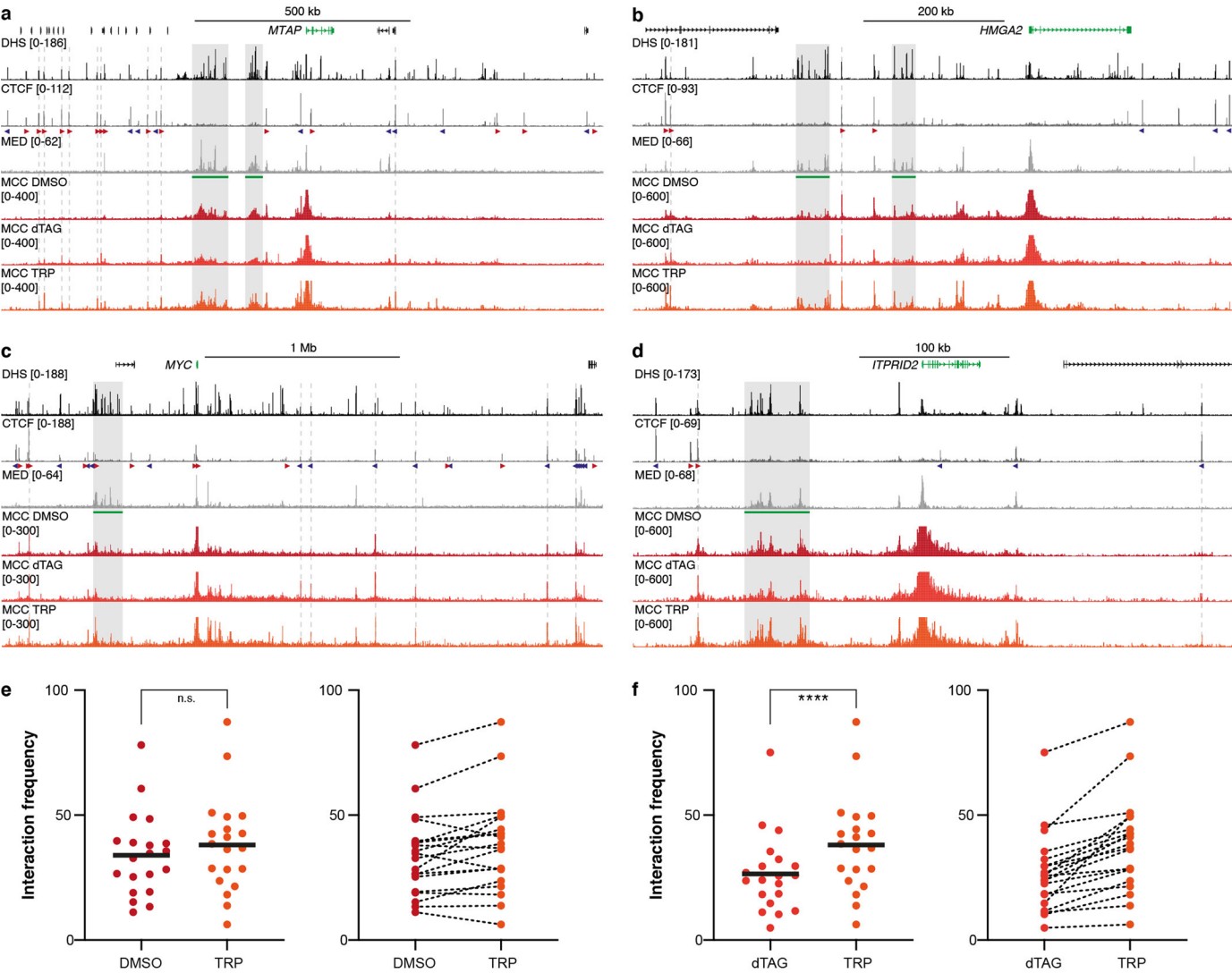

**Extended Data Fig. 9 | Micro-Capture-C analysis in HCT-116 MED14-dTAG cells treated with triptolide. a**. Micro-Capture-C (MCC) interaction profiles from the viewpoint of the *MTAP* promoter in HCT-116 MED14-dTAG cells treated with DMSO (dark red; n = 3 biologically independent samples), dTAG ligand (light red; n = 3 biologically independent samples), or triptolide (TRP; orange; n = 3 biologically independent samples). Gene annotation, DNase hypersensitive sites (DHS) and ChIP-seq data for CTCF and MED26 are shown above. Super-enhancers are highlighted in green below the MED26 profiles and orientations of CTCF motifs are indicated with arrowheads (forward orientation in red; reverse orientation in blue). The grey boxes highlight reduced interactions between the promoter and super-enhancers following dTAG treatment; the grey dashed lines highlight increased interactions with CTCF-binding sites following dTAG and triptolide treatment. The axes of the DHS and ChIP-seq profiles are scaled to signal; the axes of the MCC profiles are fixed (ranges indicated in square brackets). Coordinates (hg38): chr9:21,096,000-22,491,000. **b**. Data as described in **a** for the *HMGA2* locus. Coordinates (hg38): chr12:65,260,000-66,115,000. **c**. Data as described in **a** for the *MYC* locus. Coordinates (hg38): chr8:126,735,000-129,820,000. **d**. Data as described in **a** for the *ITPRID2* locus. Coordinates (hg38): chr2:181,700,000-182,100,000. **e**. Quantification of interaction frequencies between gene promoters and enhancer clusters extracted from MCC data in 20 loci in DMSO- and triptolide-treated cells. n.s = not significant ($p$ = 0.1155; two-sided ratio paired t-test). **f**. Quantification of interaction frequencies between gene promoters and enhancer clusters extracted from MCC data in 20 loci in dTAG- and triptolide-treated cells. **** $p$ = 3.636e-6 (two-sided ratio paired t-test).

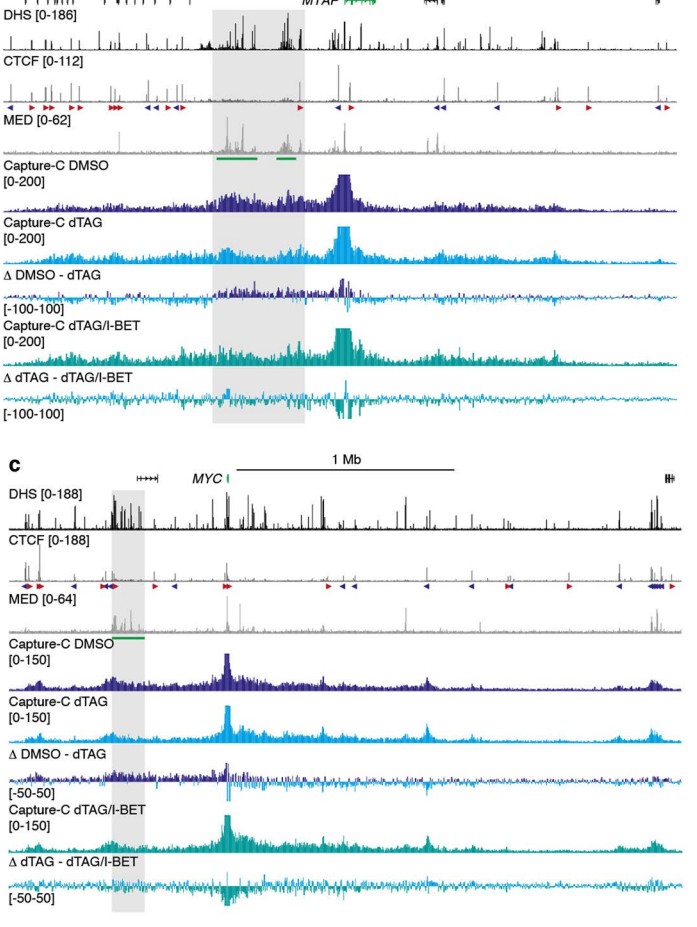
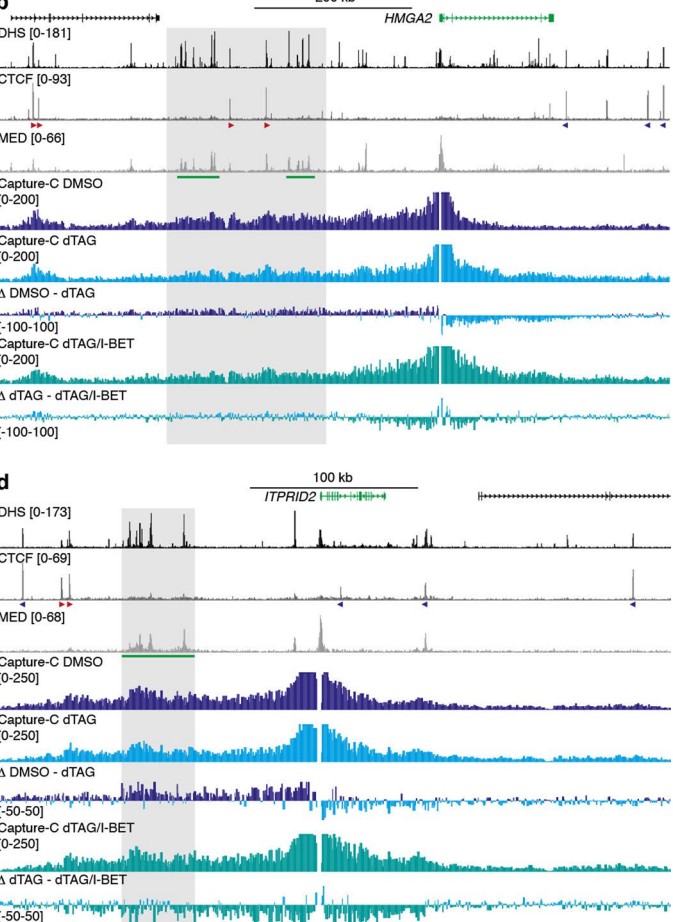

**Extended Data Fig. 10 | Capture-C analysis in HCT-116 MED14-dTAG cells treated with a BET inhibitor. a**. Capture-C interaction profiles from the viewpoint of the *MTAP* promoter in HCT-116 MED14-dTAG cells treated with DMSO (dark blue; n = 3 biologically independent samples), dTAG ligand (light blue; n = 3 biologically independent samples), or both dTAG ligand and the BET inhibitor I-BET (teal; n = 3 biologically independent samples). Gene annotation, DNase hypersensitive sites (DHS) and ChIP-seq data for CTCF and MED26 are shown above and a differential profile (Δ DMSO - dTAG) is shown below. Super-enhancers are highlighted in green below the MED26 profiles and orientations of

CTCF motifs are indicated with arrowheads (forward orientation in red; reverse orientation in blue). The axes of the DHS and ChIP-seq profiles are scaled to signal; the axes of the Capture-C profiles are fixed (ranges indicated in square brackets). Coordinates (hg38): chr9:21,096,000-22,491,000. **b**. Data as described in **a** for the *HMGA2* locus. Coordinates (hg38): chr12:65,260,000-66,115,000. **c**. Data as described in **a** for the *MYC* locus. Coordinates (hg38): chr8:126,735,000-129,820,000. **d**. Data as described in **a** for the *ITPRID2* locus. Coordinates (hg38): chr2:181,700,000-182,100,000.

# Reporting Summary

## Statistics

For all statistical analyses, confirm that the following items are present in the figure legend, table legend, main text, or Methods section.

| n/a | Confirmed | |
|---|---|---|
| ☐ | ☒ | The exact sample size ($n$) for each experimental group/condition, given as a discrete number and unit of measurement |
| ☐ | ☒ | A statement on whether measurements were taken from distinct samples or whether the same sample was measured repeatedly |
| ☐ | ☒ | The statistical test(s) used AND whether they are one- or two-sided *Only common tests should be described solely by name; describe more complex techniques in the Methods section.* |
| ☒ | ☐ | A description of all covariates tested |
| ☐ | ☒ | A description of any assumptions or corrections, such as tests of normality and adjustment for multiple comparisons |
| ☐ | ☒ | A full description of the statistical parameters including central tendency (e.g. means) or other basic estimates (e.g. regression coefficient) AND variation (e.g. standard deviation) or associated estimates of uncertainty (e.g. confidence intervals) |
| ☐ | ☒ | For null hypothesis testing, the test statistic (e.g. $F$, $t$, $r$) with confidence intervals, effect sizes, degrees of freedom and $P$ value noted *Give P values as exact values whenever suitable.* |
| ☒ | ☐ | For Bayesian analysis, information on the choice of priors and Markov chain Monte Carlo settings |
| ☒ | ☐ | For hierarchical and complex designs, identification of the appropriate level for tests and full reporting of outcomes |
| ☒ | ☐ | Estimates of effect sizes (e.g. Cohen's $d$, Pearson's $r$), indicating how they were calculated |

*Our web collection on statistics for biologists contains articles on many of the points above.*

## Software and code

Policy information about availability of computer code

| | |
|---|---|
| Data collection | Illumina NextSeq 550. |
| Data analysis | CapCruncher pipeline v.1 (https://github.com/sims-lab/CapCruncher); MCC pipeline v.1 (https://github.com/jojdavies/Micro-Capture-C), based on scripts available for academic use through the Oxford University Innovation software store (https://process.innovation.ox.ac.uk/software/p/16529a/micro- capture-c-academic/1); Bowtie2 v.2.3.5; HiC-Pro v.2.11.1; oligo design tool v.0.1.1b (https://oligo.readthedocs.io/en/latest/); Samtools v.1.9; MACS2 v.2.1.2; deepTools v.3.0.1; DiffBind v.3.6.5; DESeq2 v.1.36.0; NGseqBasic pipeline v.1 (https://github.com/Hughes-Genome-Group/NGseqBasic/releases). |

For manuscripts utilizing custom algorithms or software that are central to the research but not yet described in published literature, software must be made available to editors and reviewers. We strongly encourage code deposition in a community repository (e.g. GitHub). See the Nature Portfolio guidelines for submitting code & software for further information.

## Data

Policy information about availability of data

All manuscripts must include a data availability statement. This statement should provide the following information, where applicable:
- Accession codes, unique identifiers, or web links for publicly available datasets
- A description of any restrictions on data availability
- For clinical datasets or third party data, please ensure that the statement adheres to our policy

All raw and processed sequencing data generated in this study are available from the Gene Expression Omnibus (GEO) as a SuperSeries under accession number GSE205984. DNase-I hypersensitivity data and ChIP-Seq data for CTCF are available from ENCODE under accession codes ENCSR000ENM and ENCSR000BSE, respectively. ChIP-seq data for MED26 are available from GEO under accession code GSE121355. TT-seq data are available from GEO under access code GSE139468.

## Human research participants

Policy information about studies involving human research participants and Sex and Gender in Research.

| | |
|---|---|
| Reporting on sex and gender | Not applicable. |
| Population characteristics | Not applicable. |
| Recruitment | Not applicable. |
| Ethics oversight | Not applicable. |

Note that full information on the approval of the study protocol must also be provided in the manuscript.

# Field-specific reporting

Please select the one below that is the best fit for your research. If you are not sure, read the appropriate sections before making your selection.

☒ Life sciences          ☐ Behavioural & social sciences          ☐ Ecological, evolutionary & environmental sciences

For a reference copy of the document with all sections, see nature.com/documents/nr-reporting-summary-flat.pdf

# Life sciences study design

All studies must disclose on these points even when the disclosure is negative.

| | |
|---|---|
| Sample size | The data presented in the manuscript represent the averages of three biological replicates. These sample sizes were chosen to generate data at sufficient depth and assess differences between conditions robustly. These sample sizes are sufficient, since the observed biological effects of interest are clearly detectable between conditions and robust across replicates. For Micro-Capture-C experiments, multiple technical replicates for each biological replicate were included to boost the complexity of the data. |
| Data exclusions | No data were excluded. |
| Replication | All experiments based on sequencing data were performed for n=3 biologically independent samples as described and all attempts were successful. Immunoblots were performed independently 3 times with similar results. |
| Randomization | Samples were randomly allocated into different experimental groups prior to their treatment with dTAG ligand or DMSO. |
| Blinding | All samples were analyzed with the same pipelines, in which results are generated by scripts without interference of the researchers. Since potential expectations of the researchers cannot influence the data analysis and results, blinding is not relevant to this study. |

# Reporting for specific materials, systems and methods

We require information from authors about some types of materials, experimental systems and methods used in many studies. Here, indicate whether each material, system or method listed is relevant to your study. If you are not sure if a list item applies to your research, read the appropriate section before selecting a response.

## Materials & experimental systems

| n/a | Involved in the study |
|---|---|
| ☐ | ☒ Antibodies |
| ☐ | ☒ Eukaryotic cell lines |
| ☒ | ☐ Palaeontology and archaeology |
| ☒ | ☐ Animals and other organisms |
| ☒ | ☐ Clinical data |
| ☒ | ☐ Dual use research of concern |

## Methods

| n/a | Involved in the study |
|---|---|
| ☐ | ☒ ChIP-seq |
| ☒ | ☐ Flow cytometry |
| ☒ | ☐ MRI-based neuroimaging |

# Antibodies

| | |
|---|---|
| Antibodies used | Rabbit anti-HA-Tag (C29F4) antibody (1:1000 (Immunoblotting), 1 ug (ChIP-seq), Cell Signaling Technology, 3724), Mouse anti-GAPDH (6C5) antibody (1:2000, Abcam, ab8245), Rabbit HRP anti-Histone H3 antibody (1:5000, Abcam, ab21054), Goat anti-Rabbit IgG H&L antibody (HRP) (1:3000, Abcam, ab205718), Goat anti-Mouse IgG H&L antibody (HRP) (1:3000, Abcam, ab205719), Drosophila spike-in antibody (1 ug, Active Motif, 61686), Rabbit anti-SMC1A antibody (1:50, Abcam, ab9262), Guinea pig anti-rabbit secondary antibody (1:100, Active Motif, 53160), Rabbit IgG isotype control antibody (1:50, Cell Signaling Technology, 2729S). |
| Validation | Validation was performed by the manufacturer. The antibodies were purified using immunogen affinity and validated by immunoprecipitation, immunohistochemical analysis, and western blotting. |

# Eukaryotic cell lines

Policy information about cell lines and Sex and Gender in Research

| | |
|---|---|
| Cell line source(s) | Wild type and MED14-dTAG human colorectal carcinoma HCT-116 cells were a gift from Georg Winter (CeMM, Vienna). The MED14-dTAG HCT-116 cells were generated in Jaeger et al, Nature Genetics 2020. Wild type HCT-116 cells were originally obtained from ATCC (CCL-247). |
| Authentication | The cells were authenticated using the KaryoStat+ assay (Thermo Fisher). |
| Mycoplasma contamination | All cell lines tested negative for mycoplasma contamination. |
| Commonly misidentified lines (See ICLAC register) | No commonly misidentified lines were used. |

# ChIP-seq

## Data deposition

☒ Confirm that both raw and final processed data have been deposited in a public database such as GEO.

☒ Confirm that you have deposited or provided access to graph files (e.g. BED files) for the called peaks.

| | |
|---|---|
| Data access links *May remain private before publication.* | Cut&Tag (GSE205905) and ChIP-seq (GSE225294) data from this study are available from the Gene Expression Omnibus (GEO) under GSE205984 SuperSeries. |
| Files in database submission | GSM6235280 Cut&Tag DMSO_1<br>GSM6235281 Cut&Tag DMSO_2<br>GSM6235282 Cut&Tag DMSO_3<br>GSM6235283 Cut&Tag DMSO_4<br>GSM6235284 Cut&Tag DMSO_5<br>GSM6235285 Cut&Tag dTAG_1<br>GSM6235286 Cut&Tag dTAG_2<br>GSM6235287 Cut&Tag dTAG_3<br>GSM6235288 Cut&Tag dTAG_4<br>GSM6235289 Cut&Tag dTAG_5<br>GSM6235290 Cut&Tag IgG_DMSO<br>GSM6235291 Cut&Tag IgG_dTAG<br>GSM7043684 ChIP-seq DMSO_1<br>GSM7043685 ChIP-seq DMSO_2<br>GSM7043686 ChIP-seq DMSO_3<br>GSM7043687 ChIP-seq dTAG_1<br>GSM7043688 ChIP-seq dTAG_2<br>GSM7043689 ChIP-seq dTAG_3<br>GSM7043690 ChIP-seq Input_DMSO_1<br>GSM7043691 ChIP-seq Input_DMSO_2<br>GSM7043692 ChIP-seq Input_DMSO_3<br>GSM7043693 ChIP-seq Input_dTAG_1 |

GSM7043694 ChIP-seq Input_dTAG_2
GSM7043695 ChIP-seq Input_dTAG_3

Genome browser session
(e.g. UCSC)

SMC1A Cut&Tag:
https://genome-euro.ucsc.edu/cgi-bin/hgTracks?
db=hg38&lastVirtModeType=default&lastVirtModeExtraState=&virtModeType=default&virtMode=0&nonVirtPosition=&posit
ion=chr8%3A127403541%2D128075640&hgsid=289897223_siSABl5BvordAAzIcKzbmk8ElwPq

MED-HA ChIP-seq:
https://genome-euro.ucsc.edu/cgi-bin/hgTracks?
db=hg38&lastVirtModeType=default&lastVirtModeExtraState=&virtModeType=default&virtMode=0&nonVirtPosition=&posit
ion=chr9%3A21390261%2D21970274&hgsid=296213738_k9i9nlThJHaEvlcOQfaDaJK6pP2T

## Methodology

Replicates

Cleavage under targets and tagmentation (CUT&Tag4) experiments were performed for n=3 biologically independent samples (for a
total of 5 technical replicates) per experimental condition. ChIP-seq experiments were performed for n=3 biologically independent
samples per experimental condition.

Sequencing depth

The samples were sequenced using the NextSeq550 Illumina platform (75-bp paired-end reads) to a sequencing depth of ~10 M
reads per sample.

Antibodies

Rabbit anti-SMC1A antibody (1:50 , Abcam, ab9262); Guinea pig anti-rabbit secondary antibody (1:100, Active Motif, 53160); Rabbit
anti-HA-Tag (C29F4) antibody (1 ug, Cell Signaling Technology, 3724); Drosophila spike-in antibody (1 ug, Active Motif, 61686).

Peak calling parameters

Paired-end reads were processed for adapter removal and duplicate filtering and mapped to the hg38 reference genome using
Bowtie2. Peak calling was performed with MACS2 (consensus peaks with parameter q = 0.1 were selected). All peak profiles were
generated using Deeptools.

Data quality

The quality of the data was assessed by comparing the DMSO-treated samples to available SMC1A ChIP-seq data (Rao et al, Cell
2017) and MED26 ChIP-seq data (El Khattabi et al, Cell 2019) in HCT-116 cells.

Software

Paired-end reads were processed for adapter removal and duplicate filtering and mapped to the hg38 reference genome using
Bowtie2. Peak calling was performed with MACS2. All peak profiles were generated using Deeptools.

