## [Peer Review File · Nature Structural & Molecular Biology]

Peer Review Information

Manuscript Title: The Mediator complex regulates enhancer-promoter interactions

Corresponding author name(s): Marieke Oudelaar

Reviewer Comments & Decisions:

Decision Letter, initial version:

Message: 14th Nov 2022

Dear Dr. Oudelaar,

Thank you again for submitting your manuscript "The Mediator complex regulates enhancer-promoter interactions". I sincerely apologize for the unusual delay in responding, which resulted from the difficulty in obtaining suitable referee reports, together with our editorial team having been short-staffed in the last months. Nevertheless, we now have comments (below) from the 2 reviewers who evaluated your paper. In light of those reports, we remain interested in your study and would like to see your response to the comments of the referees, in the form of a revised manuscript.

You will see that while both reviewers find the results interesting and the data compelling, Reviewer #1 requests strengthening the statistical analysis in the SMC1A and the Cut&Tag experiments, and Reviewer #2 asks for validation of the absence of Mediator at Med14-depleted cells directly at the loci being considered. Please be sure to address/respond to all concerns of the referees in full in a point-by-point response and highlight all changes in the revised manuscript text file. If you have comments that are intended for editors only, please include those in a separate cover letter.

We expect to see your revised manuscript within 6 weeks. If you cannot send it within this time, please contact us to discuss an extension; we would still consider your revision, provided that no similar work has been accepted for publication at NSMB or published elsewhere.

As you already know, we put great emphasis on ensuring that the methods and statistics reported in our papers are correct and accurate. As such, if there are any changes that

should be reported, please submit an updated version of the Reporting Summary along with your revision.

Reporting Summary:

Please note that all key data shown in the main figures as cropped gels or blots should be presented in uncropped form, with molecular weight markers. These data can be aggregated into a single supplementary figure item. While these data can be displayed in a relatively informal style, they must refer back to the relevant figures. These data should be submitted with the final revision, as source data, prior to acceptance, but you may want to start putting it together at this point.

Data availability: this journal strongly supports public availability of data. All data used in accepted papers should be available via a public data repository, or alternatively, as Supplementary Information. If data can only be shared on request, please explain why in your Data Availability Statement, and also in the correspondence with your editor. Please note that for some data types, deposition in a public repository is mandatory - more information on our data deposition policies and available repositories can be found below:

<https://www.nature.com/nature-research/editorial-policies/reporting-standards#availability-of-data>

[Redacted]

Sincerely,
Sara

Sara Osman, Ph.D.
Associate Editor
Nature Structural & Molecular Biology

Referee expertise:

Referee #1: Epigenomics, 3D chromatin organization

Referee #2: Transcription, functional genomics

Reviewers' Comments:

Reviewer #1:

Remarks to the Author:

Ramasamy et al. present an interesting and focused study of the role of Mediator in regulating enhancer-promoter interactions. They use a combination of Capture based approaches for interrogating 3D genome structure coupled with degrons targeting mediator to show that, at least in some instances, the loss of Mediator leads to a loss of enhancer-promoter looping. This is particularly noteworthy as there has been conflicting evidence in the past of the role of Mediator in regulating 3D genome structure. Interestingly they also observe changes in Cohesin binding as well, supporting previous evidence for an interplay between these complexes. Overall, I like this study and find the data reasonably compelling. I think that it could benefit from some additional analyses and sharpening of the language in certain places, but I think that with relatively modest revisions it should be appropriate for publication. I have divided my comments below into major and minor points.

Major points:

With regards to language, the authors in the abstract write: "our results indicate that enhancer-promoter interactions are dependent on an interplay between the mediator and cohesin complexes". I think it is hard for them to make such a general conclusion when they have focused on 20 carefully selected sites in the genome, and where some of the effects they observe (34% decrease in interactions) are subtle.

There are several points where I think the authors also need to strengthen their statistical analysis. Specifically:

- The differences in SMC1A in the Cut&Tag experiments are pretty subtle. The authors state that they see differences, but they don't seem to have any set of differentially bound SMC1A sites supported by statistical methods. They have a global version of this in Fig. 4c, but it would be nice if they can also identify specific binding sites with reduced SMC1A.
- For the nano-scale patterning section, the authors mention that certain interactions are reduced (at MTAP) but there isn't really any statistical quantification of this.

They chose genes whose transcription depends on MED14 in the depletion experiments. This of course raises the question of whether the changes they see in enhancer-promoter looping are simply a reflection of the loss of transcription, as opposed to playing a more critical regulatory role. I think it would be worthwhile to perform the Capture-MCC experiments in cells treated with transcription inhibitors to see if this results in a similar loss of enhancer-promoter contacts, which would suggest that the changes in 3D genome structure are a consequence, not a cause, of the loss of expression.

Minor points:

From the plots in Fig. 1 and Supp. 2, the interactions look like they are lost upstream of the genes but gained downstream. Is this generally true? Why would Mediator contribute to some kind of upstream/downstream bias?

The micro-C data look really nice, but can they rule out that the acute depletion of Mediator changes the accessibility to MNase, and that is the cause of the reduced interactions?

The panels in Supp. Fig. 7 are really small so the tracks are even harder to see. It would be great if they can make this bigger so it is easier to see the differences.

Reviewer #2:

Remarks to the Author:

This manuscript by Ramasamy et al. provides some interesting insights about how Mediator and Cohesin may coordinately contribute to enhancer-dependent regulation of transcription at gene promoters. The data generally support the conclusions, but there are some concerns that need to be addressed and additional experiments that will improve the study.

Main points:

1. Throughout, the authors indicate the "absence" of Mediator in MED14-depleted cells. This should be directly tested at the loci probed by the authors, perhaps through ChIP-qPCR. Others have shown that degron depletion can unevenly affect chromatin-bound factors (e.g. CTCF). It is possible that MED14 depletion is uneven and that a population of chromatin-associated Mediator remains at the loci probed in this study. This information will allow the authors to better correlate their results with Mediator occupancy, and will directly relate to the model shown in Figure 6.

2. It must be acknowledged that reduced Mediator levels will negatively affect transcription independently of any looping or a condensate effects. For example, negative affects on bursting (Tantale et al. Nat Commun 2016 12248), PIC assembly, or PIC activation that may not depend on enhancer-promoter looping per se. Related to this point, promoter-bound TFs could bind Mediator to activate transcription in the absence of enhancer-promoter loops.

Other comments:

It should be noted (intro and elsewhere) that enhancer-promoter loops are transient and not uniform in a population of cells.

In the introduction, the paper from the Weiner lab (Alexander et al. eLife 2019 41769) should be noted.

Line 134-136: This summary of the results suggests that looped interactions still persist but they change "anchor points" across the locus.

Line 148: refer to panel in Fig 2.

Line 150-152: Unclear here and this needs to be re-written. For example are all regions 22% down on average?

Line 203: the co-IP experiments mentioned are not rigorous and a more thorough analysis was described in Ebmeier et al. PNAS 2010 11283.

Line 213-216: unclear description and needs to be re-written.

Line 314: it is stated about Mediator and Cohesin that "the functional relationship between these complexes has thus far been unclear" and that's true to an extent, but the results from this study do not indicate "that Mediator stabilizes Cohesin on chromatin" as stated, but it does suggest that this is true.

In the discussion (line 342-345) it is noted that the changes observed are "distinct from changes in chromatin architecture following transcription inhibition" and that's noteworthy, but it should be noted in the results section as well, because it is a question that many readers will have and it should be noted by the authors early in the description of the results, and not just later in the discussion.

Line 363: the "absence of Mediator" is likely to be an overstatement and should be re-phrased.

Fig 1, 2, 3, 4, 5: The scale for the y-axes should be shown on the Figure, not in the legend.

Fig 1: The data are not very compelling and do little to support the conclusions. Perhaps remove or move to the supplement. If a supplemental figure is shown, a comparison in which "no change" was observed would be helpful to calibrate readers.

Fig 2a: In the text (line 166) it is stated that "strong increases" at the MTAP locus are observed but readers would benefit if you could point this out on the figure. Similar concerns for Figure 5 (line 254-255): please point out on the figure.

Fig 3: The matrices need to be more clearly labeled. It is stated that the "right matrices show two different zoomed views of which the areas are indicated by the black dashed lines" but the dashed lines do not delineate the region that is zoomed in.

Fig 3: Key areas should be highlighted in the matrices to complement the description in the main text. Currently the reader must guess as to which regions of the matrices are used to support the authors' claims.

Fig 3: The intermediate 500kb panel could be removed to simplify and allow larger display of the remaining panels.

Fig 4a, b: Please label the SE regions.

Fig 4b: Please label the y-axes in panel b similar to panel a.

Fig 5: Hard to interpret for a general reader, and even to an expert it will benefit from more annotations about key regions on the data panels

Author Rebuttal to Initial comments

Reviewers' Comments:

Reviewer #1:

Remarks to the Author:

Ramasamy et al. present an interesting and focused study of the role of Mediator in regulating enhancer-promoter interactions. They use a combination of Capture based approaches for interrogating 3D genome structure coupled with degrons targeting mediator to show that, at least in some instances, the loss of Mediator leads to a loss of enhancer-promoter looping. This is particularly noteworthy as there has been conflicting evidence in the past of the role of Mediator in regulating 3D genome structure. Interestingly they also observe changes in Cohesin binding as well, supporting previous evidence for an interplay between these complexes. Overall, I like this study and find the data reasonably compelling. I think that it could benefit from some additional analyses and sharpening of the language in certain places, but I think that with relatively modest revisions it should be appropriate for publication. I have divided my comments below into major and minor points.

We would like to thank the Reviewer for their careful evaluation of our work and for their helpful suggestions to improve our study. Following the Reviewer's feedback, we have clarified our description and interpretation of the data; added additional statistical analyses; performed Micro-Capture-C experiments in cells treated with chemical inhibitors of transcription; and performed ATAC-seq experiments to confirm that the changes in the Micro-Capture-C interaction patterns are not due to changes in chromatin accessibility following Mediator depletion. We discuss these changes in detail below.

Major points:

With regards to language, the authors in the abstract write: "our results indicate that enhancerpromoter interactions are dependent on an interplay between the mediator and cohesin complexes". I think it is hard for them to make such a general conclusion when they have focused on 20 carefully selected sites in the genome, and where some of the effects they observe (34% decrease in interactions) are subtle.

We agree with the Reviewer that this sentence is an overstatement. We have changed the sentence to: "*Our results indicate that the Mediator complex can contribute to enhancerpromoter interactions*" (lines 23-24).

There are several points where I think the authors also need to strengthen their statistical analysis. Specifically:

-The differences in SMC1A in the Cut&Tag experiments are pretty subtle. The authors state that they see differences, but they don't seem to have any set of differentially bound SMC1A sites

supported by statistical methods. They have a global version of this in Fig. 4c, but it would be nice if they can also identify specific binding sites with reduced SMC1A.

We have performed additional statistical analyses to address this point: (1) We have identified all peaks that are differentially bound by SMC1A following depletion of Mediator. We have highlighted the identified differentially bound peaks in Figure 4 and Supplementary Figure 9 and provided a list of all differentially bound peaks in Supplementary Table 1. (2) We have performed statistical analysis of the global differences of SMC1A occupancy at Mediator- and CTCF-bound sites and added these to Figure 4. We have pasted the revised Figure 4 below.

-For the nano-scale patterning section, the authors mention that certain interactions are reduced (at MTAP) but there isn't really any statistical quantification of this.

We have added statistical quantification of the differences in enhancer-promoter and enhancer-enhancer interactions in the nano-scale interaction patterns at all four loci (*MYC*, *MTAP*, *HMGA2*, and *ITPRID2*). This is included in the revised Figure 5 and Supplementary Figures 10-12 and described in the figure legends.

They chose genes whose transcription depends on MED14 in the depletion experiments. This of course raises the question of whether the changes they see in enhancer-promoter looping are simply a reflection of the loss of transcription, as opposed to playing a more critical regulatory role. I think it would be worthwhile to perform the Capture-MCC experiments in cells treated with transcription inhibitors to see if this results in a similar loss of enhancer-promoter contacts, which would suggest that the changes in 3D genome structure are a consequence, not a cause, of the loss of expression.

To address this concern, we have performed Micro-Capture-C experiments in cells treated with triptolide, which inhibits transcriptional initiation. These experiments show that there is no significant reduction in enhancer-promoter interactions following acute inhibition of transcription. This is in line with previous reports based on high-resolution Micro-C analyses (Hsieh et al. *Molecular Cell* 2020; Goel et al. *BioRxiv* 2022) and indicates that the loss of enhancer-promoter interactions is not a consequence, but likely a cause of the changes in gene expression upon depletion of Mediator.

We show these results in Supplementary Figure 13 and describe them in an additional paragraph in the main text (lines 279-298). This figure and the additional paragraph are pasted below.

"The effects of Mediator depletion are distinct from transcription inhibition

The reduction in enhancer-promoter interactions that we observe after Mediator depletion is associated with a strong decrease in gene expression. A plausible explanation for these observations is that weakening of enhancer-promoter interactions leads to lower levels of gene activity. However, it is also possible that reduced transcriptional activity leads to weakening of enhancer-promoter interactions. To get more insight into the cause-consequence relationship between regulatory interactions and transcription, we performed MCC experiments in cells treated with triptolide, which inhibits initiation of transcription (Supplementary Figure 13). Comparison of the MCC data in DMSO-treated and triptolide-treated cells shows that chemical inhibition of transcription does not lead to a reduction of enhancer-promoter interactions. In contrast, we find that enhancer-promoter interactions are significantly weaker in cells in which Mediator is depleted compared to cells in which transcription is inhibited. This indicates that enhancer-promoter interactions are dependent on Mediator and not on the process of transcription."

Notably, we observe increased interactions with CTCF-binding sites following triptolide treatment. This indicates that it is possible that the increased CTCF-mediated interactions which we detect

after Mediator depletion result from the reduced levels of transcription that are associated with loss of Mediator. This is an interesting finding and consistent with previous work which shows that the distribution of Cohesin is dependent on transcription (Busslinger et al. Nature 2017; Zhang et al. Science Advances 2021) and with two recent pre-prints that show that transcribing RNA Polymerase II can form barriers to loop extrusion (Banigan et al. BioRxiv 2022; Zhang et al. BioRxiv 2022).

We further comment on the results of the transcription inhibition experiments in the Discussion of the manuscript (lines 388-402).

Minor points:

From the plots in Fig. 1 and Supp. 2, the interactions look like they are lost upstream of the genes but gained downstream. Is this generally true? Why would Mediator contribute to some kind of upstream/downstream bias?

It is not generally true that interactions are lost upstream and gained downstream. Instead, we generally find that Capture-C interactions are broadly reduced in the regions containing superenhancers. We realize that our wording to describe the Capture-C results was confusing and gave the impression of an upstream/downstream bias. We apologize for the confusion and have changed the description of these results in the main text to clarify that the regions in which interactions are reduced contain super-enhancers (lines 129-137). We have also highlighted these regions more clearly in the figures (Figure 1, Supplementary Figure 3).

The micro-C data look really nice, but can they rule out that the acute depletion of Mediator changes the accessibility to MNase, and that is the cause of the reduced interactions?

To investigate whether the changes in the Micro-Capture-C interactions patterns are related to changes in chromatin accessibility, we performed ATAC-seq experiments in DMSO- and dTAG-treated HCT-116 MED14-dTAG cells. The ATAC-seq data show that Mediator depletion does not lead to general changes in chromatin accessibility. We have included these data in Supplementary Figure 5 and added an additional paragraph to discuss in detail why we do not think that the MCC data are biased by MNase accessibility (lines 180-194). This additional paragraph and the figure are pasted below.

"It has been suggested that MNase-based 3C data could be biased by varying accessibility to MNase digestion across regions or conditions. However, the fact that we detect a significant decrease in enhancer-promoter interactions in both the MCC and the Capture-C data, which are generated with restriction enzyme digestion, indicates that reduced enhancer-promoter interactions after depletion of Mediator are unlikely to reflect underlying changes in chromatin accessibility. In addition, the observation of both decreased and increased interactions following depletion of Mediator, with increased interactions specifically overlapping with CTCF-binding sites in a convergent orientation, indicate that it is improbable that the MCC data are skewed by

nucleosome positioning. To further demonstrate that the changes in chromatin interactions in Mediator-depleted cells are not biased by potential changes in accessibility to MNase digestion, we performed ATAC-seq experiments³³ in DMSO- and dTAG-treated HCT-116 MED14-dTAG cells (Supplementary Figure 5). These experiments show that Mediator depletion does not lead to strong changes in chromatin accessibility in our regions of interest. Together, these observations indicate that the changes in enhancer/promoter interactions detected by MCC reflect bona fide changes in chromatin architecture."

The panels in Supp. Fig. 7 are really small so the tracks are even harder to see. It would be great if they can make this bigger so it is easier to see the differences.

We have increased the size of the tracks in this figure, which are now shown in Supplementary Figure 9.

Reviewer #2:

Remarks to the Author:

This manuscript by Ramasamy et al. provides some interesting insights about how Mediator and Cohesin may coordinately contribute to enhancer-dependent regulation of transcription at gene promoters. The data generally support the conclusions, but there are some concerns that need to be addressed and additional experiments that will improve the study.

We would like to thank the Reviewer for their constructive and detailed feedback on our work, which has helped us to improve our study. Following the Reviewer's suggestions, we have added ChIP-seq data to verify efficient depletion of Mediator and we have clarified the text and figures. We discuss these changes in detail below.

Main points:

1. Throughout, the authors indicate the "absence" of Mediator in MED14-depleted cells. This should be directly tested at the loci probed by the authors, perhaps through ChIP-qPCR. Others have shown that degron depletion can unevenly affect chromatin-bound factors (e.g. CTCF). It is possible that MED14 depletion is uneven and that a population of chromatin-associated Mediator remains at the loci probed in this study. This information will allow the authors to better correlate their results with Mediator occupancy, and will directly relate to the model shown in Figure 6.

To assess the efficiency of Mediator depletion at regions of interest and genome-wide, we have performed ChIP-seq experiments for Mediator in DMSO- and dTAG-treated HCT-116 MED14-dTAG cells. The ChIP-seq data show a near-complete loss of Mediator across the genome, including the regions of interest. In addition, we performed western blots of the cytosolic, nuclear and chromatin fraction of the HCT-116 MED14-dTAG cells after treatment with DMSO and dTAG. In line with the ChIP-seq data, we observe very efficient depletion of chromatin-bound Mediator. These results are shown in Supplementary Figure 1, which we have pasted below (on the next page).

Although the ChIP and chromatin fraction western blots indicate efficient depletion of Mediator, we agree with the Reviewer that the word "absence" is likely an overstatement, and have changed this throughout the text.

2. It must be acknowledged that reduced Mediator levels will negatively affect transcription independently of any looping or a condensate effects. For example, negative affects on bursting (Tantale et al. Nat Commun 2016 12248), PIC assembly, or PIC activation that may not depend on enhancer-promoter looping per se. Related to this point, promoter-bound TFs could bind Mediator to activate transcription in the absence of enhancer-promoter loops.

We agree with the Reviewer that this is relevant to mention and have added an additional paragraph to discuss this, in which we refer to the study from Tantale et al. (lines 340-345). We have pasted this paragraph below:

"In addition, the Mediator complex plays a central role in the regulation of gene expression and is thought to act at several stages of the transcription cycle. It is therefore likely that the large decrease in transcriptional output upon Mediator depletion is not only related to weaker enhancer-

promoter interactions, but also to the loss of the general function of Mediator in the regulation of initiation (e.g. PIC assembly and activation), re-initiation, elongation, and transcriptional bursting^{22,45}."

Other comments:

It should be noted (intro and elsewhere) that enhancer-promoter loops are transient and not uniform in a population of cells.

We have added statements to highlight this, both in the Introduction (lines 31-32) and in the Discussion (lines 351-353) of the manuscript.

In the introduction, the paper from the Weiner lab (Alexander et al. eLife 2019 41769) should be noted.

We agree with the Reviewer that this paper is important to mention. We have added a brief discussion of the paper to the Discussion of the manuscript (lines 353-356), since it fits very well with the section in which we discuss the discrepancy between the impact of Mediator depletion on enhancer-promoter interaction frequency and transcription levels.

Line 134-136: This summary of the results suggests that looped interactions still persist but they change "anchor points" across the locus.

We apologize for the confusion. The Reviewer is correct and we do not claim that all "loops" are lost. Instead, we describe a specific decrease of enhancer-promoter interactions and a specific increase of CTCF interactions following depletion of Mediator. This is very clear from the Micro-Capture-C data, but also reflected in the Capture-C data. We have clarified our discussion of these results (lines 129-134).

Line 148: refer to panel in Fig 2.

We have included the panels in the reference (line 149).

Line 150-152: Unclear here and this needs to be re-written. For example are all regions 22% down on average?

We have clarified this in the text (lines 151-156) and included additional graphs to Figure 2 to further explain this point. We have pasted these graphs and the revised text below.

"We find that depletion of Mediator leads to a decrease in the frequency of enhancer-promoter interactions in the 20 regions we focused on (Supplementary Figure 4). Quantification of the MCC interactions between gene promoters and clusters of Mediator-bound enhancers indicates an average reduction of 22% across these regions (Figure 2c). The reduction in interaction frequency between the promoters and a narrow region covering the largest Mediator peak within these broad clusters is on average 34% (Figure 2d)."

Line 203: the co-IP experiments mentioned are not rigorous and a more thorough analysis was described in Ebmeier et al. PNAS 2010 11283.

We thank the Reviewer for bringing this reference to our attention and have added it to the manuscript (lines 221-223).

Line 213-216: unclear description and needs to be re-written.

We apologize for the confusion and have changed this section (lines 232-241). To further clarify the Cohesin CUT&Tag data and make it easier for readers to directly relate the CUT&Tag data to the MCC data, we have also swapped the regions shown in the main and supplementary figures. Consistent with Figures 1 and 2, Figure 4 now shows CUT&Tag data for the *MTAP* and *HMGA2* loci. The *ERRF1* and *KRT19* loci (which were shown in the previous version of Figure 4) are now shown in Supplementary Figure 9.

Line 314: it is stated about Mediator and Cohesin that "the functional relationship between these complexes has thus far been unclear" and that's true to an extent, but the results from this study do not indicate "that Mediator stabilizes Cohesin on chromatin" as stated, but it does suggest that this is true.

We agree with the Reviewer that this is an overstatement and have changed this sentence to clarify that a model in which Mediator stabilizes Cohesin on chromatin is a possible explanation for our observations which requires further validation (lines 361-367).

In the discussion (line 342-345) it is noted that the changes observed are "distinct from changes in chromatin architecture following transcription inhibition" and that's noteworthy, but it should be noted in the results section as well, because it is a question that many readers will have and it should be noted by the authors early in the description of the results, and not just later in the discussion.

This point has been raised by Reviewer 1 as well and we have added additional experiments to demonstrate that transcription inhibition does not reduce enhancer-promoter interactions. We have added a paragraph to the Results section (lines 279-298) and an additional figure (Supplementary Figure 13) to describe the transcription inhibition experiments.

Line 363: the "absence of Mediator" is likely to be an overstatement and should be re-phrased.

We agree with the Reviewer and have changed this throughout the text.

Fig 1, 2, 3, 4, 5: The scale for the y-axes should be shown on the Figure, not in the legend.

We have changed this for all main and supplementary figures.

Fig 1: The data are not very compelling and do little to support the conclusions. Perhaps remove or move to the supplement. If a supplemental figure is shown, a comparison in which "no change" was observed would be helpful to calibrate readers.

We understand the Reviewer's point about this figure and agree that it is in principle not necessary to support the main conclusions of our work. However, there are two reasons why we would prefer to keep this figure as a main figure. First, the direct comparison of the CaptureC and the Micro-Capture-C data highlights the precision of the Micro-Capture-C technique and the importance of investigating enhancer-promoter interactions with very high-resolution approaches. Since the Micro-Capture-C technique is still a very recent development, we feel that this is an interesting point to make. Second, the Capture-C data provide an orthogonal dataset that indicates that depletion of Mediator leads to loss of enhancer-promoter interactions, even though the effects detected in the Capture-C data are very subtle. We have clarified these two points in the main text (lines 158-162 and lines 180-185, respectively). In addition, we have added extra Capture-C experiments for two loci at which gene expression levels do not change after Mediator depletion for comparison (Supplementary Figure 3e-f).

Fig 2a: In the text (line 166) it is stated that "strong increases" at the MTAP locus are observed but readers would benefit if you could point this out on the figure. Similar concerns for Figure 5 (line 254-255): please point out on the figure.

We have clarified Figure 2 and 5 to highlight the changes in interactions after Mediator depletion.

Fig 3: The matrices need to be more clearly labeled. It is stated that the "right matrices show two different zoomed views of which the areas are indicated by the black dashed lines" but the dashed lines do not delineate the region that is zoomed in.

Fig 3: Key areas should be highlighted in the matrices to complement the description in the main text. Currently the reader must guess as to which regions of the matrices are used to support the authors' claims.

Fig 3: The intermediate 500kb panel could be removed to simplify and allow larger display of the remaining panels.

We have simplified and clarified this figure according to the Reviewer's suggestions. We have labelled the matrices more clearly and added a black box to delineate the region for which we provide a zoomed in view. We have also highlighted areas of interest in the matrices and removed the intermediate zoom panel.

Fig 4a, b: Please label the SE regions.

Fig 4b: Please label the y-axes in panel b similar to panel a.

We have clarified this figure following the Reviewer's suggestions. The super-enhancers are indicated with a green bar and we have changed the labels of the y-axes. As suggested by Reviewer 1, we have also indicated significant changes with grey highlights.

Fig 5: Hard to interpret for a general reader, and even to an expert it will benefit from more annotations about key regions on the data panels.

We have clarified this figure and added highlights to indicate the regions of interest in which interaction patterns change.

Decision Letter, first revision:

Message: Our ref: NSMB-A46618A

9th Mar 2023

Dear Dr. Oudelaar,

Thank you for submitting your revised manuscript "The Mediator complex regulates enhancer-promoter interactions" (NSMB-A46618A). It has now been seen by the original referees and their comments are below. The reviewers find that the paper has improved in revision, and therefore we'll be happy in principle to publish it in Nature Structural & Molecular Biology, pending minor revisions to satisfy the referees' final requests and to

comply with our editorial and formatting guidelines.

We are now performing detailed checks on your paper and will send you a checklist detailing our editorial and formatting requirements in a couple of weeks. Please do not upload the final materials and make any revisions until you receive this additional information from us.

To facilitate our work at this stage, it is important that we have a copy of the main text as a word file. If you could please send along a word version of this file as soon as possible, we would greatly appreciate it; please make sure to copy the NSMB account (cc'ed above).

Sincerely,
Sara

Sara Osman, Ph.D.
Associate Editor
Nature Structural & Molecular Biology

Reviewer #1 (Remarks to the Author):

The authors have satisfied all of my criticisms. Congratulations on their interesting study.

Reviewer #2 (Remarks to the Author):

I thank the authors for their responses, which have clarified some points and have improved the study. I have no additional concerns.

Author Rebuttal, first revision:

Reviewer #1:

Remarks to the Author:

The authors have satisfied all of my criticisms. Congratulations on their interesting study.

We are happy that our revisions have satisfied the Reviewer's criticisms and would like to thank the Reviewer for their constructive feedback.

Reviewer #2:

Remarks to the Author:

I thank the authors for their responses, which have clarified some points and have improved the study. I have no additional concerns.

We are happy that our revisions have clarified the points raised by the Reviewer and would like to thank the Reviewer for their constructive feedback.

Final Decision Letter:**Message** 30th May 2023

:

Dear Dr. Oudelaar,

We are now happy to accept your revised paper "The Mediator complex regulates enhancer-promoter interactions" for publication as an Article in Nature Structural & Molecular Biology.

Your paper will be published online soon after we receive proof corrections and will appear in print in the next available issue. You can find out your date of online publication by contacting the production team shortly after sending your proof corrections. Content is published online weekly on Mondays and Thursdays, and the embargo is set at 16:00 London time (GMT)/11:00 am US Eastern time (EST) on the day of publication. Now is the time to inform your Public Relations or Press Office about your paper, as they might be interested in promoting its publication. This will allow them time to prepare an accurate and satisfactory press release. Include your manuscript tracking number (NSMB-A46618B) and our journal name, which they will need when they contact our press office.

About one week before your paper is published online, we shall be distributing a press release to news organizations worldwide, which may very well include details of your work. We are happy for your institution or funding agency to prepare its own press release, but it must mention the embargo date and Nature Structural & Molecular Biology. If you or your Press Office have any enquiries in the meantime, please contact press@nature.com.

Please note that *Nature Structural & Molecular Biology* is a Transformative Journal (TJ). Authors may publish their research with us through the traditional subscription access route or make their paper immediately open access through payment of an article-processing charge (APC). Authors will not be required to make a final decision about access to their article until it has been accepted. Find out more about Transformative Journals <https://www.springernature.com/gp/open-research/transformative-journals>

Authors may need to take specific actions to achieve [compliance](https://www.springernature.com/gp/open-research/funding/policy-compliance-faqs) with funder and institutional open access mandates. If your research is supported by a funder that requires immediate open access

(e.g. according to [Plan S principles](https://www.springernature.com/gp/open-research/plan-s-compliance)) then you should select the gold OA route, and we will direct you to the compliant route where possible. For authors selecting the subscription publication route, the journal's standard licensing terms will need to be accepted, including [self-archiving policies](https://www.springernature.com/gp/open-research/policies/journal-policies). Those licensing terms will supersede any other terms that the author or any third party may assert apply to any version of the manuscript.

Sincerely,
Sara

Sara Osman, Ph.D.
Associate Editor
Nature Structural & Molecular Biology
